# Causality Inspired Federated Learning for OOD Generalization

Jiayuan Zhang [1]  Xuefeng Liu [1 2]  Jianwei Niu [1 2]  Shaojie Tang [3]  Haotian Yang [1]  Xinghao Wu [1]

## Abstract

The out-of-distribution (OOD) generalization problem in federated learning (FL) has recently attracted significant research interest. A common approach, derived from centralized learning, is to extract causal features which exhibit causal relationships with the label. However, in FL, the global feature extractor typically captures only invariant causal features shared across clients and thus discards many other causal features that are potentially useful for OOD generalization. To address this problem, we propose FedUni, a simple yet effective architecture trained to extract all possible causal features from any input. FedUni consists of a comprehensive feature extractor, designed to identify a union of all causal feature types in the input, followed by a feature compressor, which discards potential *inactive* causal features. With this architecture, FedUni can benefit from collaborative training in FL while avoiding the cost of model aggregation (i.e., extracting only invariant features). In addition, to further enhance the feature extractor's ability to capture causal features, FedUni add a causal intervention module on the client side, which employs a counterfactual generator to generate counterfactual examples that simulate distributions shifts. Extensive experiments and theoretical analysis demonstrate that our method significantly improves OOD generalization performance.

## 1. Introduction

*Federated Learning* (FL) has emerged as a privacy-preserving framework for collaborative learning across distributed clients. A practical problem, how to enable the FL model to robustly generalize to target clients with unknown distributions, referred to as the *out-of-distribution (OOD) generalization* problem, has recently attracted significant research attention. Unfortunately, deep learning models are prone to relying on non-causal relationships (i.e., shortcuts) for prediction (Geirhos et al., 2020), leading to significant performance degradation when the test distributions fall outside the training scope. For instance, in Fig. 1, 'dogs' and 'grass' exhibit high statistical dependence in the training dataset of *Client A*, which can easily mislead the model into making predictions based on the grass background (non-causal features) rather than the shape of dogs (causal features). Recent works in centralized learning scenarios have proposed addressing the OOD generalization problem through the lens of causality, focusing on learning causal features that maintain an invariant causal relationship to the output label across various data distributions. It is widely recognized that causal features are far more robust to distribution shifts compared to non-causal ones.

However, in FL, the global feature extractor is typically trained to extract causal features that remain invariant across all clients (Zhang et al., 2021; Nguyen et al., 2022; Tang et al., 2022; Guo et al., 2023b; Liao et al., 2024). Although these invariant features are generally robust and reliable, many valuable causal features that are potentially useful for OOD generalization are inevitably discarded. This limitation becomes more pronounced in the presence of data heterogeneity, where only a small subset of invariant causal features are shared across clients. As an illustrative example, consider the task of learning a global model for dog classification in FL, as shown in Fig. 1. In this setup, *Client A* has real-world photos of dogs, while *Client B*'s dataset consists of dog sketches. The global feature extractor, trained to capture invariance across clients, preserves only invariant causal features like shape while discarding client-specific ones such as details of the dog's mouth and tail. When the test data distribution matches that of *Client A*'s training data, discarding these detailed causal features can lead to suboptimal generalization. In contrast, preserving all causal features enables clients to flexibly select those most relevant

[1]State Key Laboratory of Virtual Reality Technology and Systems, School of Computer Science and Engineering, Beihang University [2]Zhongguancun Laboratory, Beijing, China [3]Department of Management Science and Systems, School of Management, Center for AI Business Innovation, University at Buffalo, Buffalo, NY, USA. Correspondence to: Jianwei Niu <niujianwei@buaa.edu.cn>, Xuefeng Liu <liu_xuefeng@buaa.edu.cn>.

*Proceedings of the $42^{nd}$ International Conference on Machine Learning*, Vancouver, Canada. PMLR 267, 2025. Copyright 2025 by the author(s).

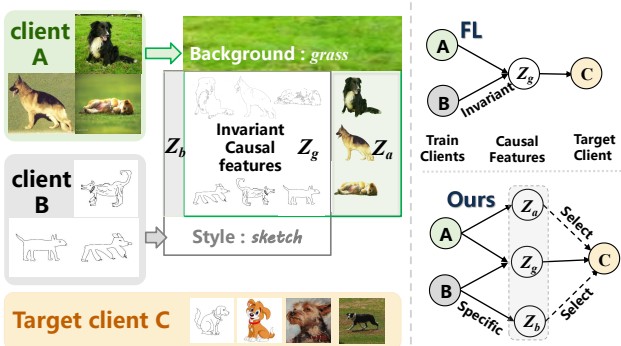

*Figure 1.* Taking dog classification as an example to illustrate FedUni. FedUni preserves all causal features present across clients, rather than only the invariant causal features, thereby achieving better generalization to unseen target clients.

to their target domain, thereby improving generalization performance. For instance, the causal features of samples from the cartoon domain may constitute a combination of features found in both real-world photos and sketches.

To address the above limitation, we propose that the global feature extractor in FL should capture the full set of causal features present across all clients, rather than only the invariant subset shared by all. However, this objective is challenging because different clients possess varying quantities and types of causal features. Additionally, not all causal features are necessarily beneficial for generalization to a given target client. For example, in Fig. 1, the causal features extracted by the global feature extractor contain all causal features from both *Client A* and *Client B*. However, when deployed to a target client whose data distribution is more similar to *Client B*'s sketch style, the detailed features of dogs become irrelevant to the target domain. Retaining these irrelevant causal features may even degrade generalization performance.

To overcome this challenge, we introduce a simple yet effective architecture, named FedUni, which consists of two key components: a comprehensive feature extractor followed by a feature compressor. The former is designed to extract the union of all causal feature types across clients. Notably, if a given input lacks a particular type of causal feature, the corresponding feature is identified as an *inactive* causal feature. The feature compressor is then employed to effectively discard these *inactive* causal features, mitigating their potential negative impact on the model's performance. These two components are trained simultaneously, so that the feature extractor can benefit from model aggregation while avoiding the extraction of only invariant features. In addition, to further enhance the feature extractor's ability to capture causal features and eliminate non-causal ones, we incorporate a causal intervention module on the client side. This module employs a counterfactual generator to transform images

into counterfactual examples that simulate distribution shift between source and target domains while preserving the semantic information. The main contributions of this work are outlined as follows:

- We formulate a structural causal model (SCM) for the OOD generalization problem in FL and propose that the global feature extractor should retain all causal features rather than limiting itself to only the invariant ones shared across clients.

- We introduce an effective method, FedUni, which incorporates a comprehensive feature extractor to identify the union of all causal feature types in the input. This is followed by a feature compressor to discard potential *inactive* causal features. To further enhance the ability to capture causal features, FedUni employs a counterfactual generator for causal intervention.

- Extensive experiments and theoretical analysis demonstrate that our method significantly improves OOD generalization performance.

## 2. Related Work

**Causal Feature Learning.** One promising approach to addressing OOD generalization is leveraging causal features underlying the observed data (Zhou et al., 2022). A series of works derived from Invariant Risk Minimization (IRM) (Arjovsky et al., 2019) formulate invariant causal mechanisms by incorporating information theory and nonlinear prediction functions (Ahuja et al., 2021; Krueger et al., 2021; Yang et al., 2023). More closely related to our method, (Chen et al., 2024) reveals that existing methods may still learn fake causal features and introduces conditional mutual information to rectify them. Another line of research generates counterfactual samples and enforces consistency of causal features between original and augmented samples, as shown in (Chang et al., 2021; Lv et al., 2022; Noohdani et al., 2024). However, these methods focus on centralized scenarios where all training data is accessible.

**Federated Learning for OOD Generalization.** The OOD generalization problem in FL has gained significant research interest. Existing methods extract invariant features via adversarial domain alignment (Zhang et al., 2021; Qi et al., 2025; 2023; 2024; Meng et al., 2024), feature regularization (Nguyen et al., 2022), or inter-client gradient alignment (Guo et al., 2023b; Wu et al., 2022). However, they primarily focus on inter-client invariance, potentially discarding valuable client-specific information beneficial for OOD generalization. Meanwhile, FedSDR (Tang et al., 2023) and FedPIN (Tang et al., 2024) aim to preserve personalized causal features. However, these methods are designed for personalized federated learning (PFL) (Tang et al., 2021;

2022), where the causal features of each training client are assumed to be known. Our problem is more challenging as the types and quantities of causal features for a target client are unseen during training. Retaining only invariant features may discard valuable ones, whereas preserving all causal features could introduce training-specific causal features that are irrelevant to the target distribution, potentially degrading OOD generalization performance.

A more detailed discussion of related work is provided in Appendix A.

## 3. Problem Formulation

**Notations.** Let $\mathcal{X}$, $\mathcal{Y}$ and $\mathcal{E}$ represent the input, target and environment space respectively. Suppose there are $N$ clients, and the local data $D_{e_c}^c$ on the $c$-th client, which contains $n_c$ samples, is drawn from the training environment $\mathcal{E}_{tr}$ ($e_c \in \mathcal{E}_{tr}$). For convenience, following (Guo et al., 2023b), we formalize the model as $f = w \circ \phi$, where $\phi : \mathcal{X} \rightarrow \mathcal{Z} \subset \mathbb{R}^n$ is the feature extractor that maps the input space $\mathcal{X}$ to the feature representation space $\mathcal{Z}$ and $w : \mathcal{Z} \rightarrow \hat{\mathcal{Y}}$ is the classifier. The overall model is parameterized as $f_\theta(\cdot) = w(\phi(\cdot))$, where $\theta = (w, \phi)$. In the standard FL paradigm, the goal is to learn a global model that minimizes the average expected risk over all participating clients. The global expected empirical loss is denoted as $\mathcal{R}(f) := \sum_{c=1}^{N} \mathbb{E}_{(x,y) \in D_{e_c}^c}[l(f_\theta(x), y)]$, where $l$ is the loss function.

### 3.1. FL OOD Generalization

The goal of FL OOD generalization is to learn a global model $f_\theta$ with $\mathbb{D} = \{D_{e_c}^c \mid c \in [1, N], c \in \mathbb{Z}\}$ to enable prediction on samples drawn from arbitrary environments in $\mathcal{E}$. This objective can be formulated as a min-max optimization problem as follows:

$$\mathcal{R}_\mathcal{E}(f) := \max_{e \sim \mathcal{E}} \min_{\theta} [\mathcal{R}^e(f_\theta)]. \tag{1}$$

Obviously, the optimization problem in Eq. (1) cannot be solved directly, as only the training environment $\mathcal{E}_{tr}$ instead of $\mathcal{E}$ is accessible during training. A common assumption in FL OOD generalization is that there exist latent causal variables (features) $Z_C \in \mathbb{R}^d$, which possess the following environment-invariant property (Arjovsky et al., 2019):

$$\mathbb{P}_\theta(Y|Z_C, E = e) = \mathbb{P}_\theta(Y|Z_C, E = e'), \forall e, e' \in \mathcal{E}. \tag{2}$$

To enable the global feature extractor $f$ to extract $Z_C$, FL methods extract inter-client invariant features from distributions (Liu et al., 2021), representations (Nguyen et al., 2022), and gradients (Guo et al., 2023b). However, these methods focus only on the invariance of causal features, without considering their completeness. Retaining only an invariant subset of causal features may discard useful information for OOD generalization, leading to suboptimal

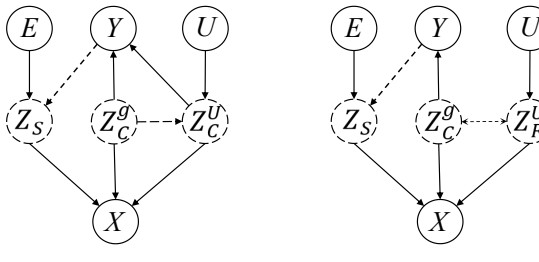

(a) Existing SCM      (b) Restructured SCM

*Figure 2.* SCMs for FL OOD generalization. (The solid circle represents *observed variables*, the dashed circle represents *latent variables*, the solid arrow indicates *causal dependency*, and the dashed arrow represents *Markov dependency*.)

generalization performance. We propose to solve the OOD generalization problem in FL through the lens of causality.

### 3.2. A Causality Viewpoint to FL OOD Generalization

It is known that OOD generalization is possible under practical causal assumptions (Ahuja et al., 2021).

**Existing SCM.** Recently, some works have formulated SCM in FL considering the distribution heterogeneity among clients (Tang et al., 2023; 2024), as shown in Fig. 2a, where $E$ is the **E**nvironment variables, and $U \in \{1, \cdots, N\}$ is the **U**ser/Client indicator. $Z_S$ denotes non-causal variables. The causal variables are divided into two components: the shared/global causal variables $Z_C^g$ and the personalized/local causal variables $Z_C^U$. The personalized causal variables $Z_C^U$ include both $Z_C^g$ and additional invariant information derived from $U$.

**Assumption 3.1** (Existing SCM illustrated in Fig. 2a)**.**

$$Z_S := f_{spu}(E, Y), \ Z_C^U := f_{inv}(U, Z_C^g),$$
$$X := f_{gen}(Z_S, Z_C^g, Z_C^U), \ Y := \omega(Z_C^g, Z_C^U).$$

According to Assumption 3.1, existing methods assume that the global model should extract invariant causal variables $Z_C^g = \left\{ \bigcap_{u=1}^N Z_C^{U=u} \right\}$ shared across clients. However, as shown by Lemma 3.2, this objective may lead to the extracted invariant features being incomplete.

**Lemma 3.2.** *If the data generating process of each client follows the SCM assumption illustrated in Fig. 2a, the global invariance $Z_C^g$ is incomplete from two perspectives:*

- $Y \not\perp U \mid Z_C^g$: *Under $Z_C^g$, the prediction of $Y$ is still influenced by the client indicator $U$.*

- $Y \perp [U, E] \mid Z_C^g, Z_C^U$: *Given $Z_C^g$ and $Z_C^U$, $Y$ is conditionally independent of the environment variable $E$ and user indicator $U$.*

The first claim in Lemma 3.2 suggests that the global model may exhibit inconsistent performance across different ID/OOD data distributions. The second claim indicates that combining $Z_C^g$ and $Z_C^U$ can further reduce the empirical risk, and that $Z_C^U$ should be retained for better inference performance. This indicates that existing SCMs are not well suited for the OOD generalization problem in FL, where the global model is expected to exhibit fairness and robust adaptability across heterogeneous data distributions.

**Restructured SCM.** To better address the FL OOD generalization problem, we propose a reconstructed SCM, as shown in Fig. 2b. A key distinction in our formulation lies in the definition of the global causal variable $Z_C^g$. Unlike existing approaches that define $Z_C^g$ as the intersection of client-specific causal variables, we define $Z_C^g = \left\{ \bigcup_{u=1}^{N} Z_C^{U=u} \right\}$ as the union of $Z_C^U$ across all clients for any given input. In addition, since the global causal variable may contain information that is not required for the current task, we introduce $Z_F^U$ to represent these ***Inactive causal features***, which are causal features specific to certain distributions but non-causal for the target task. Formally, we define this as $Z_F^U = Z_C^g / Z_C^U$.

**Assumption 3.3** (Our Restructured SCM illustrated in Fig. 2b)**.**

$Z_S := f_{spu}(E, Y),\ Z_C^g := f_{inv}(Z_F^U),\ Z_F^U := f_{inv}(U, Z_C^g),$
$X := f_{gen}(Z_S, Z_C^g, Z_F^U),\ Y := \omega(Z_C^g).$

**Lemma 3.4.** *If the data generating process of each client follows the SCM assumption illustrated in Fig. 2b, then the following three statements hold:*

- *$Z_S \perp [Z_C^g, Z_F^U] \mid Y$: Given $Y$, both the $Z_C^g$ and $Z_F^U$ are conditionally independent of $Z_S$.*

- *$Y \perp [U, E] \mid Z_C^g$: Considering that $Z_F^U \subsetneq Z_C^g$, components in $[U, E]$ are conditionally independent of the target $Y$ given $Z_C^g$ and $Z_F^U$.*

- *$Y \not\perp Z_C^g | Z_S, Z_F^U$: $Y$ is not conditionally independent of $Z_C^g$ given $Z_S$ and $Z_F^U$.*

Lemma 3.4 highlights the potential for designing a better FL method for OOD generalization. Based on the first claim, the non-causal features can be directly excluded from the causal ones, and the second claim indicates that the global invariant features $Z_C^g$ can be extracted via collaborative learning among clients. Moreover, the third claim indicates that we should reconsider the Markov dependency between $Z_C^g$ and $Z_F^U$, and these variables need to be distinguished based on their differing roles in label prediction when conditioned on each other. All proofs related to this section are provided in Appendix B.

## 4. Methodology

**Overview.** Inspired by the restructured SCM discussed above, we propose a novel FL paradigm named FedUni, consisting of three components: (1) a causal intervention module on the client side to distinguish causal features $Z_C^g$ from non-causal ones $Z_S$ (Section 4.1), (2) an inactive causal feature compressor that discards *inactive* causal features $Z_F^U$ from all causal features $Z_C^g$ specific to the data distribution (Section 4.2), and (3) A comprehensive feature extractor designed to capture the union of all causal feature types present in a given input, achieved by training it jointly with the other two modules. The comprehensive feature extractor and inactive causal feature compressor are uploaded for global aggregation and transferred to the target clients, while the causal intervention module remains local to the training clients. The theoretical analysis of the generalization boundary is presented in Appendix C.

### 4.1. Causal Intervention Module

We expect the FL model to extract causal features $Z_C^g$ while eliminating non-causal ones $Z_S$. Unfortunately, directly reconstructing the causal features $Z_C^g$ is impractical. Nonetheless, inspired by (Lv et al., 2022), we propose two properties that these features should satisfy.

**Proposition 4.1.** *Causal variables $Z_C^g$ should satisfy the following two properties:*

- ***Invariance under causal intervention***: *Based on the first claim in Lemma 3.4, we have $Z_C^g \perp E \mid Y$. Thus, performing an intervention upon $E$ does not make changes to $Z_C^g$.*

- ***Sufficiency of semantic information***: *The causal variable $Z_C^g$ should be causally sufficient to the label prediction, i.e. maximizing $I(Z_C^g; Y)$.*

Existing methods typically perform predefined data augmentations on samples to serve as a causal intervention module (Lv et al., 2022; Chen et al., 2023). However, these approaches lack diversity and are insufficient to simulate the distribution shifts in the real world. Inspired by recent single domain generalization (Guo et al., 2023a; Yang et al., 2024), we employ a trainable counterfactual generator $g : \mathcal{X} \to \mathcal{X}'$, parameterized by $\psi$, to generate more diverse counterfactual samples while preserving semantic information. Below, we propose the optimization objective for $g$.

**(1) Diversify non-causal variables.** To increase the diversity of generated counterfactual samples, the generated image $x'$ should have the minimal correlation with the source image $x$. Mutual information (MI), denoted as $I(x; x')$, provides a natural measure of this correlation (Wang et al.,

2021):

$$I(x; x') = \mathbb{E}_{p(x,x')}[log \frac{p(x'|x)}{p(x')}] = H(x') - H(x'|x), \quad (3)$$

where $H(x')$ is the entropy of $x'$ and $H(x'|x)$ is the conditional entropy of $x'$ given $x$. To prevent $H(x') = 0$, during optimization, we employ a normalized variant $N(x; x')$ of the mutual information, namely the *coefficient of constraint* (Press, 2007):

$$\min_{\psi} N(x; x') = \min_{\psi} \frac{I(x; x')}{H(x')} = 1 - \max_{\psi} \frac{H(x'|x)}{H(x')}, \quad (4)$$

where $N(x; x') = 0$ if and only if $x'$ and $x$ are independent. For simplicity, we use a parameter-free approximation to estimate the solution.

**Lemma 4.2.** *$H(x'|x)$ can be approximated by the conditional probability $p(x'|x)$ as $\mathcal{L}_1$-Laplacians, as follows:*

$$H(x'|x) = -\mathbb{E}_{x'}(\log P(x'|x)) \approx \mathbb{E}_x[\|x - \psi(x)\|_1].$$

*Similarly, $H(x')$ can be approximated by the marginal distribution of $x'$:*

$$H(x') = -\mathbb{E}_{x'}(\log P(x')) \approx \mathbb{E}_x[\|\psi(x)\|_1].$$

The detailed proof of the above lemma can be found in Appendix B.4. Based on this approximation, we define the following information-theoretic loss:

$$\mathcal{L}_{DIV} = \max_{\psi} \frac{H(x'|x)}{H(x')} \approx \max_{\psi} \mathbb{E}_x \frac{\|\psi(x) - x\|_1}{\|\psi(x)\|_1}. \quad (5)$$

**(2) Preserve semantic information.** To ensure that the semantic information of $x'$ and $x$ does not change drastically, we impose constraints on the unbounded generation domain.

**Lemma 4.3.** *Maximizing the diversity loss is approximately equivalent to maximizing the lower bounds of $L_{DIV}$, which satisfy the following inequality:*

$$\max_{\psi} \mathcal{L}_{DIV} \geq \max_{\psi} \mathbb{E}_x \left( \left| 1 - \frac{\|x\|_1}{\|x'\|_1} \right| \right).$$

*It can be observed that generating the counterfactual image $x'$ essentially involves maximizing $\|\|x'\|_1 - \|x\|_1\|_p$.*

The detailed proof of this lemma can be found in Appendix B.4. Following Lemma 4.3, we constrain the distance between the generated image and the original image by adding a penalty on the $l_1$-norm of the samples:

$$\mathcal{L}_{REG} = \min_{\psi} \mathbb{E}_x[\|\|x'\|_1 - \|x\|_1\|_1]. \quad (6)$$

**Optimization.** The optimization objective for the generator is defined as follows, where $\alpha$ is a hyper-parameter used to adjust the strength of the regularization term:

$$\mathcal{L}_g = \min_{\psi} -\mathcal{L}_{DIV} + \alpha \mathcal{L}_{REG}. \quad (7)$$

### 4.2. Inactive Causal Feature Compressor

We now proceed to discuss how to compress $Z_F^U$ from $Z_C^g$. Notably, the features extracted by the global feature extractor consist of the true causal features $Z_C^U$ given the distribution $U$, the *inactive* causal features $Z_F^U$ and the non-causal parts $Z_S$, which can be denoted as:

$$\phi(X) = Z_C^U \cup Z_F^U \cup Z_S, \text{ where } Z_C^U = Z_C^g / Z_F^U. \quad (8)$$

To preserve semantic information, we need to maximize the conditional mutual information (CMI) to strengthen the causal path $Z_C^U \to Y$:

$$\max_{Z_C^U} I[Y; Z_C^U \mid Z_F^U, Z_S]. \quad (9)$$

Similarly, we observe an inactive causal dependency path between $Z_F^U$ and $Y$ ($Z_F^U \to Z_C^g \to Y$). As the label prediction $Y$ is expected to be conditionally independent of $Z_F^U$, we have $Y \perp Z_F^U \mid Z_C^U, Z_S$. To compress the *inactive* causal features, we propose minimizing the CMI between $Y$ and $Z_F^U$, conditioned on $Z_C^U$ and $Z_S$:

$$\min_{Z_F^U} I[Y; Z_F^U \mid Z_C^U, Z_S]. \quad (10)$$

For simplicity, we assume that after the causal intervention steps, the non-causal features $Z_S$ have been removed, i.e., $\phi^*(X) = Z_C^U \cup Z_F^U$ and $Z_C^U = \phi^*(X)/Z_F^U$. Then, Eq. (9) and Eq. (10) can be simplified to:

$$\min_{Z_F^U} \lambda I[Y; Z_F^U \mid \phi^*(X)/Z_F^U] - I[Y; \phi^*(X)/Z_F^U \mid Z_F^U]. \quad (11)$$

**Lemma 4.4.** *The CMI admits the following decompositions:*

$$I[Y; Z_F^U \mid \phi^*(X)/Z_F^U] = -H(Y|\phi^*(X)) + H(Y|\phi^*(X)/Z_F^U),$$
$$I[Y; \phi^*(X)/Z_F^U \mid Z_F^U] = -H(Y|\phi^*(X)) + H(Y|Z_F^U).$$

The proof of this lemma is provided in Appendix B.5. According to Lemma 4.4, Eq. (11) can be decomposed as:

$$\min_{Z_F^U} H(Y|\phi^*(X)/Z_F^U) - \lambda H(Y|Z_F^U). \quad (12)$$

The objective implies that we only need to compress features from $\phi^*(X)$ to strengthen the causal dependency path $Z_C^U \to Y$ and discourage the inactive causal path $Z_F^U \to Z_C^g \to Y$. We propose an inactive causal feature compressor $s : \mathcal{Z} \to \mathcal{Z}'$, which is a simple two-layer architecture. The compressor $s(\cdot)$ receives the features extracted by the global feature extractor $\phi(\cdot)$ and performs soft feature selection to compress $Z_F^U$.

**Optimization.** The optimization objective of the inactive causal feature compressor in Eq. (11) can be expressed as follows:

$$\mathcal{L}_s = \min_{s, \omega} \mathbb{E}_x[l(\omega(s(\phi(x)) \odot \phi(x)), y)] - \\ \lambda \mathbb{E}_x[l(\omega(1 - s(\phi(x))) \odot \phi(x), y)]. \quad (13)$$

where $\lambda$ is a hyper-parameter and $\odot$ denotes the element-wise product and $(1 - s(\phi(x))) \odot \phi(x)$ represents the compressed part $Z_F^U$.

---

**Algorithm 1** FedUni

---

1: **Input:** $T, K, N, \eta, \omega_g^0(\Phi_g^0), s_g^0$
2: **for** $t = 0$ **to** $T - 1$ **do**
3:     Randomly select a client subset $S_t$ from $N$ clients.
4:     Broadcast $\omega_g^0(\Phi_g^0), s_g^0$ to selected clients.
5:     **for** each client $c \in S_t$ **do**
6:         **for** local steps $k$ **to** $K - 1$ **do**
7:             Update counterfactual generator:
8:                 $\psi_c^t = \psi_c^t - \eta \nabla L_g$
9:             Update compressive global model:
10:                $\omega_u^t, \Phi_u^t = \omega_u^t, \Phi_u^t - \eta \nabla L_f$
11:             Update feature compressor:
12:                $s_u^t, \omega_u^t = s_u^t, \omega_u^t - \eta \nabla L_s$
13:         **end for**
14:     **end for**
15:     Model aggregation:
16:         $\omega_g(\Phi_g), s_g = \frac{1}{|S_t|} \sum_c \omega_u^t(\Phi_u^t), \frac{1}{|S_t|} \sum_c s_u^t$
17: **end for**
18: **Return:** global model $\omega_g(\Phi_g), s_g$

---

### 4.3. Comprehensive Feature Extractor

The goal of the comprehensive feature extractor is to extract all causal features from any input across all clients, which can be achieved by training it simultaneously with the two aforementioned modules.

Firstly, with the causal intervention module, $Z_S$ can be eliminated by minimizing the causal effect of environmental changes. We use the contrastive loss (Chen et al., 2020) as the objective to enforce the consistency of features between the original image $x$ and the counterfactual image $x'$:

$$\mathcal{L}_{CI} = \mathbb{E}_x \log \frac{e^{d(\phi(x),\phi(x'))/\tau}}{e^{d(\phi(x),\phi(x'))/\tau} + \sum_{X' \in B'} e^{d(\phi(x),\phi(X'))/\tau}}, \tag{14}$$

where the original and counterfactual scenes of the same instance are treated as a positive pair, while the features drawn from other counterfactual samples are considered negative pairs. Moreover, the causal features should be sufficient for the label prediction. We use the ERM loss (Vapnik, 2013) to optimize the model to learn the semantic information related to target $Y$:

$$L_{SEM} = R(f) + \mathbb{E}_x l(f_\theta(g_\psi(x)), y). \tag{15}$$

Secondly, with the inactive causal compressor module, we require that unselected features $Z_F^U$ still retain some semantically relevant information:

$$\mathcal{L}_{ADV} = \min_{\phi,\omega} \mathbb{E}_x[l(\omega((1 - s(\phi(x))) \odot \phi(x)), y)]. \tag{16}$$

This encourages the feature extractor to capture richer causal features. Meanwhile, this can prevent certain feature locations from being consistently ignored, which could otherwise lead to the inclusion of non-informative features.

**Optimization.** The overall optimization objective can be written in the following form, where $\beta$ is used to control the proportion of the causal intervention loss in the training objective:

$$L_f = \min_{\phi,\omega} \mathcal{L}_{SEM} + \mathcal{L}_{ADV} + \beta \mathcal{L}_{CI}. \tag{17}$$

The detailed optimization procedure is outlined in Algorithm 1.

## 5. Experiments

### 5.1. Experimental Settings

**Datasets.** To evaluate the effectiveness of our method, we conduct experiments on both the **spurious correlation** dataset, i.e. Waterbirds (Sagawa et al., 2019), Colored-MNIST/FMNIST (Arjovsky et al., 2019; Ahuja et al., 2020) and the **cross-domain** datasets, i.e., Digits, PACS (Li et al., 2017).

**Configurations.** For Waterbirds and PACS, we use the ResNet-18 (He et al., 2016) pretrained on ImageNet (Deng et al., 2009) as backbone. With regard to CMNIST and CFMNIST, an MLP with one hidden layer serves as the model. The AlexNet (Deng et al., 2009) is selected for Digits dataset. Unless otherwise mentioned, the local update step is $5$ and the mini-batch size is $64$. The learning rate $lr = 1.41 \times 10^{-4}$. In our setting, $\alpha, \beta, \lambda$ are set to $0.1, 1, 0.01$ respectively. The test results are based on the model that performs best on the validation set sampled from the training data with a sampling ratio of $0.2$. For more detailed experimental settings, please refer to Appendix D.

**Comparison.** We compare our method with 9 state-of-the-art algorithms: FedAvg (McMahan et al., 2017) which is a classic FL method; three methods for addressing heterogeneity (FedProx (Li et al., 2020), Scaffold (Karimireddy et al., 2020), MOON (Li et al., 2021b)); and four FL generalization methods (FedSR (Nguyen et al., 2022), FedIIR (Guo et al., 2023b), FedDG-GA (Zhang et al., 2023), FedSDR (Tang et al., 2023)). It is worth noting that FedSDR is designed for Personalized Federated Learning (PFL); in our experiments, we use a variant that combines FedSDR with FedAvg.

### 5.2. Overall Performance

**Effect of spurious correlations.** To validate the effectiveness of the proposed method in eliminating the impact of spurious correlations and thereby enhancing OOD generalization performance, we evaluate the test accuracy of the

global model on a range of diverse test data distributions on Waterbirds, CMNIST, and CFMNIST. The worst-case ('ID-worst' and 'OOD-worst') and average-case ('ID-avg' and 'OOD-avg') accuracy are summarized in Table 1. Due to the preservation of richer causal information during training, our method demonstrates superior performance on both ID and OOD data distributions compared to other SOTA methods.

**Effect of domain shift.** Compared with spurious correlation datasets, the non-causal features (i.e. style) are tightly coupled with the causal features of images, making them more difficult to remove. Moreover, the data distributions across different domains exhibit stronger heterogeneity. As shown in Table 2, our method demonstrates significantly better generalization performance across almost all domains, with reduced variance in cross-domain generalization.

### 5.3. Verification

**Analysis of causal and *inactive* causal features.** FedUni decouples the features extracted by the comprehensive feature extractor into two parts, causal features $Z_C^U$ and *inactive* causal features $Z_F^U$. Fig. 3 presents the T-SNE (Van der Maaten & Hinton, 2008) visualization of causal and *inactive* causal features on the Digits dataset, where the first row represents causal features and the second row represents *inactive* causal features. The visualization demonstrates that causal features exhibit higher discriminative characteristics, as they have a stronger causal relationship with the label compared to *inactive* causal features. To provide a quantitative evaluation, we further compute the Fisher Score (Gu et al., 2012) in Fig. 4, which measures feature discriminability by assessing the ratio of inter-class variance to intra-class variance, formulated as $F = \sum_i n_i(\mu_i - \mu)^2 / \sum_i n_i \sigma_i^2$, where $n_i$, $\mu_i$, and $\sigma_i^2$ represent the number of samples, mean, and variance of the $i$-th class, respectively. In addition, it is worth noting that the gap in Fisher Scores between causal and *inactive* causal features increases as the dataset complexity rises. This indicates that more comprehensive causal features are extracted from complex datasets, which contain a higher proportion of *inactive* causal features.

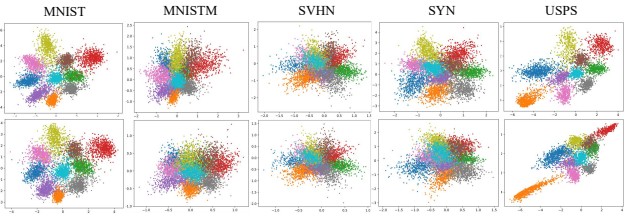

*Figure 3.* T-SNE visualization of causal and *inactive* causal features on the Digits dataset.

**Analysis of overlapped ratio of causal features.** To analyze the overlap of selected causal features among different

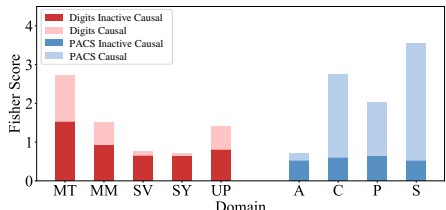

*Figure 4.* Fisher Scores of causal and *inactive* causal features on Digits and PACS datasets.

distributions, the *inactive* causal feature compressor generates a mask $s(\phi(x))$ for each given data distribution, and the mean of these masks is used as the overall causal features types for the current data distribution. Fig. 5a shows the cosine similarity of feature selection masks among training clients, grouped by dataset origins: MNISTM (1-2), SVHN (3-4), SYN (5-6), and USPS (7-10). It can be observed that clients from the same domain exhibit a higher overlap in the selection of causal features. We further analyze the similarity in causal feature selection between test data and the training clients, as shown in Fig. 5b. It is found that domains present in the training data retain feature selection consistency within their respective groups. For the unseen test domain MNIST, its feature selection overlaps more with MNISTM, which shares similar shape, and USPS, which shares the same black and white color scheme. These findings further validate the motivation behind FedUni, which is designed to select task-relevant causal features for different data distributions. Furthermore, domains with higher similarity tend to exhibit more consistent causal feature selection.

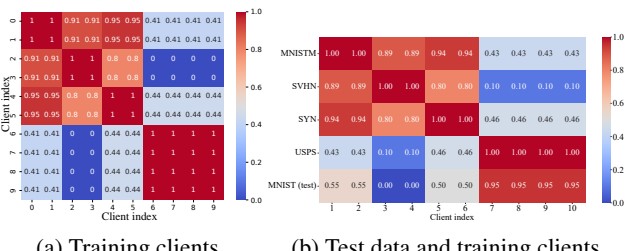

(a) Training clients.  (b) Test data and training clients

*Figure 5.* Overlapped ratios of causal features.

**Analysis of causal features redundancy.** We now investigate how different components of our setup affect the learned features. In this section, we compare 5 different methods, which are: (1) **Base**: FedAvg using ERM update locally; (2) **+CI**: Incorporating a causal intervention module to the base model; (3) **+SL**: Incorporating inactive causal feature compressor to the base model; (4) **+CI+SL**: Incorporating both of the aforementioned modules and using the features extracted by the feature extractor for prediction; (5) **+CI+SL(s)**: Based on (4), using the features after the

| Datasets | Waterbirds | | | | CMNIST | | | | CFMNIST | | | |
|---|---|---|---|---|---|---|---|---|---|---|---|---|
| Test Acc (%) | ID-avg | ID-worst | OOD-avg | OOD-worst | ID-avg | ID-worst | OOD-avg | OOD-worst | ID-avg | ID-worst | OOD-avg | OOD-worst |
| FedAvg | 87.11 | 86.30 | 79.09 | 71.26 | 94.11 | 85.52 | 68.53 | 27.63 | 92.55 | 64.41 | 87.48 | 44.52 |
| FedProx | 87.31 | 86.41 | 78.80 | 70.42 | 94.32 | 85.42 | 68.23 | 26.10 | 94.43 | 74.43 | 86.40 | 36.82 |
| Scaffold | 87.41 | 86.20 | 78.41 | 70.95 | 93.56 | 84.54 | 68.47 | 29.16 | 94.01 | 71.41 | 86.70 | 41.10 |
| MOON | 85.42 | 84.69 | 79.25 | 73.27 | 93.77 | 83.15 | 73.92 | 39.62 | 93.60 | 69.20 | 86.64 | 39.71 |
| FedSR | 87.27 | 86.55 | 79.86 | 74.08 | 95.19 | 85.70 | 72.69 | 37.41 | 92.87 | 67.22 | 86.60 | 43.12 |
| FedIIR | 87.12 | 86.14 | 78.67 | 70.56 | 93.99 | 84.91 | 68.99 | 28.69 | 92.92 | 65.72 | 87.08 | 43.21 |
| FedDG-GA | 86.50 | 85.64 | 79.21 | 72.16 | 94.68 | 88.33 | 69.02 | 26.90 | 92.58 | 64.32 | 87.22 | 44.13 |
| FedSDR | 86.93 | 86.22 | 80.60 | 74.59 | 94.44 | 83.71 | 68.72 | 27.81 | **94.53** | 74.10 | 86.23 | 40.52 |
| Ours | **88.92** | **88.38** | **83.29** | **78.05** | **95.87** | **89.53** | **88.03** | **56.41** | 94.43 | **74.54** | **88.70** | **46.25** |

*Table 1.* Performance comparison of different methods on Waterbirds, CMNIST and CFMNIST datasets. (Bolded values represent the best performance in the experiment setting, while underlined values indicate the second-best performance)

| Datasets | Digits | | | | | | PACS | | | | |
|---|---|---|---|---|---|---|---|---|---|---|---|
| Test Acc (%) | MNIST | MNISTM | SVHN | SYN | USPS | Avg. | Art | Cartoon | Photo | Sketch | Avg. |
| FedAvg | $82.38_{\pm1.31}$ | $34.69_{\pm0.69}$ | $22.62_{\pm0.16}$ | $41.07_{\pm0.71}$ | $80.51_{\pm1.16}$ | $52.25_{\pm0.46}$ | $66.27_{\pm1.09}$ | $61.07_{\pm0.66}$ | $86.76_{\pm0.76}$ | $47.69_{\pm1.07}$ | $65.44_{\pm0.21}$ |
| FedProx | $81.54_{\pm2.37}$ | $34.57_{\pm0.55}$ | $21.77_{\pm0.91}$ | $38.00_{\pm0.79}$ | $77.20_{\pm0.94}$ | $50.62_{\pm0.77}$ | $70.69_{\pm0.93}$ | $64.43_{\pm2.88}$ | $84.36_{\pm1.69}$ | $49.51_{\pm0.41}$ | $67.25_{\pm0.08}$ |
| Scaffold | $85.92_{\pm0.94}$ | $35.09_{\pm0.74}$ | $22.772_{\pm0.10}$ | $41.06_{\pm0.90}$ | $79.82_{\pm1.27}$ | $52.93_{\pm0.41}$ | $66.49_{\pm0.77}$ | $61.16_{\pm0.56}$ | $87.42_{\pm1.20}$ | $49.36_{\pm1.63}$ | $66.11_{\pm0.39}$ |
| MOON | $82.85_{\pm0.94}$ | $34.81_{\pm0.72}$ | $22.62_{\pm0.42}$ | $41.81_{\pm0.73}$ | $78.12_{\pm1.09}$ | $52.05_{\pm0.56}$ | $71.07_{\pm0.58}$ | $66.51_{\pm1.02}$ | $87.38_{\pm1.44}$ | $43.11_{\pm1.17}$ | $67.02_{\pm0.53}$ |
| FedSR | $88.94_{\pm0.34}$ | $36.46_{\pm0.66}$ | $25.23_{\pm0.12}$ | $48.09_{\pm1.19}$ | $87.50_{\pm0.78}$ | $57.24_{\pm0.18}$ | $70.53_{\pm2.17}$ | $64.49_{\pm1.84}$ | $88.18_{\pm0.31}$ | $51.64_{\pm2.16}$ | $68.71_{\pm1.07}$ |
| FedIIR | $86.27_{\pm0.79}$ | $35.87_{\pm1.35}$ | $22.61_{\pm0.30}$ | $44.03_{\pm1.26}$ | $81.43_{\pm3.10}$ | $54.04_{\pm1.19}$ | $70.71_{\pm0.30}$ | $62.89_{\pm0.26}$ | $87.33_{\pm0.82}$ | $48.48_{\pm0.40}$ | $67.35_{\pm0.44}$ |
| FedDG-GA | $85.54_{\pm0.57}$ | $34.49_{\pm0.53}$ | $23.55_{\pm0.85}$ | $40.12_{\pm0.79}$ | $79.54_{\pm1.78}$ | $52.65_{\pm0.38}$ | $67.84_{\pm1.44}$ | $64.96_{\pm0.85}$ | $90.40_{\pm0.68}$ | $48.11_{\pm0.51}$ | $67.11_{\pm0.20}$ |
| FedSDR | $89.72_{\pm0.50}$ | $33.74_{\pm0.86}$ | $22.98_{\pm2.00}$ | $44.73_{\pm1.27}$ | $88.45_{\pm0.56}$ | $55.92_{\pm0.40}$ | $70.71_{\pm0.23}$ | $66.76_{\pm0.72}$ | $87.51_{\pm1.20}$ | $49.58_{\pm0.74}$ | $69.36_{\pm0.41}$ |
| Ours | **$89.85_{\pm0.97}$** | **$52.40_{\pm1.81}$** | **$35.42_{\pm2.45}$** | **$66.29_{\pm0.95}$** | **$90.48_{\pm0.95}$** | **$66.85_{\pm0.64}$** | **$71.08_{\pm0.65}$** | **$68.76_{\pm1.51}$** | $89.07_{\pm0.89}$ | **$76.04_{\pm1.14}$** | **$76.24_{\pm0.60}$** |

*Table 2.* Performance comparison of different methods on Digits and PACS datasets.

inactive causal feature compressor for prediction.

We calculate the redundancy of features on the Digits dataset using the formula: $R = \frac{1}{d^2} \sum_i \sum_j |\rho(X_i, X_j)|$, where $\rho(X_i, X_j)$ is the Pearson correlation between a pair of feature dimensions $i$ and $j$. As shown in Fig. 6, both the compressor and the causal intervention module can improve test accuracy as well as feature redundancy. The observed low test accuracy and feature redundancy in the base model suggest that it has extracted non-causal features (i.e., shortcuts) for prediction, while discarding diverse causal features. The incorporation of the compressor retains client-specific causal features, while the causal intervention module eliminates spurious correlations. Together, they better preserve the underlying causal features. However, not all causal features contribute meaningfully to the test task-some are *inactive*. Filtering features through the compressor allows the model to discard redundant or inactive features, thereby improving both redundancy metrics and predictive performance.

### 5.4. Ablation Study

**Ablation Study on Model Components.** We conduct ablation studies to evaluate the individual and combined contributions of FedUni components, as shown in Table 3. Both the generator and compressor individually enhance performance compared to their absence. Notably, their combination achieves the best performance across all domains,

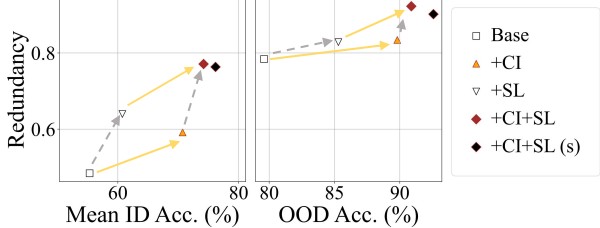

*Figure 6.* The relationship between feature redundancy and ID/OOD test accuracy. Gray and Orange arrows denote changes due to adding feature compressor and causal intervention module, respectively.

demonstrating the effectiveness of integrating both components.

| Components | | Test Accuracy (%) | | | | | |
|---|---|---|---|---|---|---|---|
| $-\mathcal{L}_{DIV} + \mathcal{L}_{REG}$ | $\mathcal{L}_s$ | MNIST | MNISTM | SVHN | SYN | USPS | Avg. |
| ✗ | ✗ | 82.39 (1.31) | 34.69 (0.69) | 22.62 (0.16) | 41.07 (0.71) | 80.51 (1.16) | 52.26 (0.46) |
| ✓ | ✗ | 86.57 (0.47) | 49.20 (2.13) | 34.08 (0.72) | 59.11 (1.04) | 87.89 (0.37) | 63.37 (0.67) |
| ✗ | ✓ | 88.19 (0.31) | 36.81 (0.24) | 23.90 (0.38) | 48.67 (0.32) | 87.44 (0.08) | 57.00 (0.14) |
| ✓ | ✓ | **88.65** (0.97) | **52.40** (1.81) | **35.42** (2.45) | **66.29** (0.98) | **90.48** (0.95) | **66.65** (0.64) |

*Table 3.* Ablation study on the generator and filter module on Digits dataset. (The gray background represents the variance of repeated experiments.)

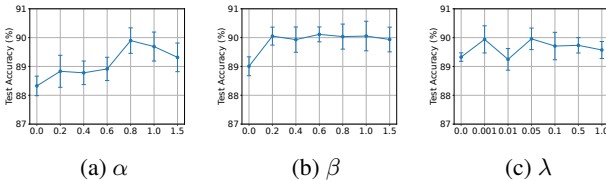

*Figure 7.* The ablation of hyper-parameters on CMNIST.

**Hyper-parameter Sensitivity.** The proposed method involves three hyper-parameters $\alpha$, $\beta$, and $\lambda$, where $\alpha$ represents the degree of the regularization term in Eq. (7), $\beta$ corresponds to the proportion of the causal intervention loss in the training objective Eq. (17), and $\lambda$ denotes the selection ratio of the soft-feature filter in Eq. (12). We conduct experiments on the CMNIST dataset and performed 5 repeated runs. We vary one hyper-parameter while keeping the other two fixed. The results are shown in Fig. 7. It can be observed that our approach is robust to hyper-parameter changes, with a maximum fluctuation of 2.57%. This indicates that FedUni does not rely heavily on careful hyper-parameter tuning and can maintain stable performance under a wide range of configurations.

## 6. Conclusion

In this work, we address the OOD generalization challenge in FL by proposing FedUni, a novel framework that retains all possible causal features across clients rather than only an invariant subset. Our approach leverages a comprehensive feature extractor and a feature compressor to extract the union of causal features while suppressing *inactive* causal features. Additionally, we introduce a causal intervention module that employs counterfactual generation to further enhance OOD generalization. Through extensive theoretical analysis and empirical evaluation, we demonstrate that FedUni significantly improves OOD generalization. These findings highlight the importance of preserving diverse causal features in FL.

## Acknowledgment

This work was supported by the National Natural Science Foundation of China under Grants 62372028 and 62372027.

## Impact Statement

This paper presents work whose goal is to advance the field of Machine Learning. There are many potential societal consequences of our work, none of which we feel must be specifically highlighted here.

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

# A. Related Work

**Causal Feature Learning.** An important research direction to tackle OOD generalization is to exploit the causal features behind the observed data (Zhou et al., 2022). A series of works formulate invariant causal mechanisms in terms of features. Invariant risk minimization (IRM) (Arjovsky et al., 2019) provides a solution for learning causal features, but it dramatically deteriorates when applied to an over-parameterized model. From this perspective, some works (Ahuja et al., 2021; Krueger et al., 2021) further extend the IRM by incorporating information theory and nonlinear prediction functions. IIB (Li et al., 2022) minimizes invariant risks for nonlinear classifiers through a tractable mutual information-based loss function. CaSN (Yang et al., 2023) introduces the classical concept of the Probability of Necessity and Sufficiency (PNS) to simultaneously capture information on both sufficient and necessary causes and proposes PNS risk to optimize OOD generalization capability. Another series of papers achieves this goal by generating counterfactual images and constraining the consistency between the features before and after generation (Chang et al., 2021). CIRL (Lv et al., 2022) introduce Fourier transformation as a causal intervention module to enforce features to be separable from non-causal factors, jointly independent, and causally sufficient for classification. DaC (Noohdani et al., 2024) leverages the model's Class Activation Map (CAM) to identify causal components in images, then intervenes by composing different elements to generate counterfactual samples and retrains the model for improved generalization. However, these works focus on centralized scenarios where all training data is accessible.

**Single Domain Generalization** is a more challenging yet practical domain generalization task, where the training data comes from only a single distribution. A typical line of methods synthesize diverse samples from hypothetical domains and ensure consistency between features extracted from the source and synthetic samples. Among the earliest attempts, ADA (Volpi et al., 2018) and M-ADA (Qiao et al., 2020) introduce an adversarial data augmentation strategy to tackle the worst-case domain generalization problem. Recent studies inspired by style transfer methods (Gatys et al., 2016) focus on generating samples with novel styles. To name a few, PDEN (Li et al., 2021a) proposes a progressive domain expansion framework. L2D (Wang et al., 2021) presents a style-complement module designed to synthesize images through the optimization of the mutual information's upper bound. UDP (Guo et al., 2023a) directly augments images using in the image space, decoupling the training objectives between the generator and the discriminator, thereby avoiding mutual interference. Despite their remarkable effectiveness, these methods often introduce additional computational burdens. In this paper, we take into account the limited computational capabilities of clients in FL and devises a more lightweight algorithm.

**General Federated Learning (GFL).** The classic FedAvg (McMahan et al., 2017) algorithm performs well on IID datasets, but its performance significantly drops under the presence of data heterogeneity. One line of methods has been proposed to mitigate the impact of data heterogeneity during the client-side training process (Zhang et al., 2022; Mendieta et al., 2022; Zhu et al., 2024b; Wu et al., 2024; Zhu et al., 2022; Wu et al., 2023; 2025; Zhu et al., 2024b). For example, FedProx (Li et al., 2020) introduces an *L2-norm* term between the global model parameters and the local parameters. SCAFFOLD (Karimireddy et al., 2020) adds correction terms for local gradients to ensure that the local update moves towards the true optimum. MOON (Li et al., 2021b) propose a model-level contrastive learning approach inspired by conventional data-level contrastive frameworks. Another line of studies focus on enhancing the server-side aggregation process. FedNova (Wang et al., 2020) enhances model aggregation by addressing heterogeneous local steps caused by varying data quantities or training epochs. FedDNA (Duan et al., 2021) selectively apply importance-weighted adversarial learning to batch normalization layers while maintaining standard FedAvg for gradient-based parameters. In the context of server-side optimization (Sharma et al., 2022), FedAvgM (Hsu et al., 2019) implements momentum-accelerated SGD. While these methods comprehensively address distribution shifts among participating clients, it critically overlooks the generalization requirements of non-participating clients. Unlike these works, our method attempts to learn latent causal relationships that work equally well for both the participating and non-participating clients.

**Federated Learning for OOD Generalization.** The out-of-distribution (OOD) generalization problem in FL has recently attracted significant research interest (Zhang et al., 2024; Zhu et al., 2024a). Data augmentation-based research involves applying various transformations to the existing training data (Yoon et al., 2021; Hao et al., 2021). Several approaches enhance OOD generalization by optimizing the weighting strategies for global model aggregation. FedDG-GA (Zhang et al., 2023) dynamically calibrates aggregation weights through generalization adjustment to achieve tighter theoretical bounds. More similar to our approach, several studies concentrate on invariant (causal) feature extraction through adversarial domain alignment or causal representation learning. FedADG (Zhang et al., 2021) introduces a federated adversarial domain adaptation framework to align data distributions across all clients. FedSR (Nguyen et al., 2022) imposes dual regularization mechanisms on latent representations and conditional mutual information, suppressing spurious correlations while preserving invariant features. FL GAME (Zhang et al., 2021) establishes a multi-agent game-theoretic architecture

enabling decentralized parallel computation through Nash equilibrium-driven optimization, which systematically extracts client-invariant causal features via information bottleneck constraints. FedIIR (Guo et al., 2023b) implicitly learns causal relationships for OOD generalization by quantifying disagreement across clients and reducing it through inter-client gradient alignment in parameter space. However, these methods focus on mining the invariant features across clients, which inevitably leads to the loss of some valuable information.

**Causal Feature with FL OOD Generalization.** Recently, inspired by causal feature learning in centralized learning scenarios, a growing number of methods aim to extract causal features to improve the OOD generalization ability of FL model (Tang et al., 2021; 2022). FedSDR(Tang et al., 2023) constructs a structural causal model (SCM) for heterogeneous clients to collaboratively identify non-causal features and learn the optimal personalized causally invariant predictor. FedPIN (Tang et al., 2024) distinguishes personalized causal features from non-causal ones using global invariant features as anchors, formulating an information-theoretic constraint to enable shortcut-averse personalized invariant learning, thereby improving OOD generalization in PFL. However, these methods are especially designed for PFL, where the causal features of each client's data are observable during training. In contrast, in the GFL setting we focus on, the causal features present in the unknown test data are unseen during training.

## B. Theoretical Guarantees

### B.1. The detailed explanation of Assumption 3.1 and the proof of Lemma 3.2

**Assumption 3.1** (Existing SCM illustrated in Fig. 2a)

$$Z_S := f_{spu}(E, Y), \ Z_C^U := f_{inv}(U, Z_C^g),$$
$$X := f_{gen}(Z_S, Z_C^g, Z_C^U), \ Y := \omega(Z_C^g, Z_C^U).$$

*Explanation.* Here, $Z_S$ represents spurious non-causal features. It is a function of the environment $E$ and target $Y$. $Z_C^U$ is the local causal variables, which is determined by the client indicator $U$ and the global causal features $Z_C^g$ through the invariable causal function $f_{inv}$. The data $X$ is generated by the generation function $f_{gen}$, which takes into account the spurious variables $Z_S$, the global causal variables $Z_C^g$, and the local causal variables $Z_C^U$. Finally, the target $Y$ is determined by the classifier $\omega$ based on the global causal variables $Z_C^g$ and the local causal variables $Z_C^U$.

**Lemma 3.2.** If the data generating process of each client follows the SCM assumption illustrated in Fig. 2a, the global invariance $Z_C^g$ is incomplete from two perspectives:

- $Y \not\perp U \mid Z_C^g$: Under $Z_C^g$, the prediction of $Y$ is still influenced by the client indicator $U$.

- $Y \perp [U, E] \mid Z_C^g, Z_C^U$: Given $Z_C^g$ (i.e. global/shared invariant features) and $Z_C^U$ (i.e. local/personalized invariant features), $Y$ is conditionally independent of the environment variable $E$ and user indicator $U$.

*Proof.* To prove the non-independence $Y \not\perp U \mid Z_C^g$, we rely on the d-separation criterion. Specifically, there is an unblocked causal path from $U$ to $Y$ ($U \to Z_C^U \to Y$) that does not pass through $Z_C^g$. This implies that even when conditioned on $Z_C^g$, the prediction of $Y$ is still influenced by the client indicator $U$.

To prove the conditional independence $Y \perp [U, E] \mid Z_C^g, Z_C^U$, we observe that the variables $Z_C^g$ and $Z_C^U$ d-separate $Y$ from both the environment indicator $E$ and the user/client indicator $U$. $\square$

### B.2. The detailed explanation of Assumption 3.3 and the proof of Lemma 3.4

**Assumption 3.3** (Our Restructured SCM illustrated in Fig. 2b)

$$Z_S := f_{spu}(E, Y), \ Z_C^g := f_{inv}(Z_F^U), \ Z_F^U := f_{inv}(U, Z_C^g),$$
$$X := f_{gen}(Z_S, Z_C^g, Z_F^U), \ Y := \omega(Z_C^g).$$

*Explanation.* Here, $Z_S$ represents spurious non-causal variable and is a function of the environment $E$ and the target $Y$. Different from Assumption 3.1, $Z_C^g$ is now the only cause of the target $Y$, which means it encompasses the local causal

knowledge carried by $Z_C^U$ in Assumption 3.1. We have a new variable $Z_F^U$ (where $F$ stands for False). $Z_F^U$ is only affected by the client indicator $U$ and influences $Y$ through affecting $Z_C^g$. $Z_F^U$ is a local causal variable, similar to $Z_C^U$ in Assumption 3.1, representing some causal information that holds locally at the client but my not be transferred well to all environments.

**Lemma 3.4.** If the data generating process of each client follows the SCM assumption illustrated in Fig. 2b, then the following three statements hold:

- $Z_S \perp [Z_C^g, Z_F^U] \mid Y$: Given $Y$, both the $Z_C^g$ and $Z_F^U$ are conditionally independent of the non-causal variable $Z_S$.

- $Y \perp [U, E] \mid Z_C^g$: Components in $[U, E]$ are conditionally independent of the target $Y$ given $Z_C^g$ and $Z_F^U$. Considering that $Z_F^U \subsetneq Z_C^g$, this expression is a simplified notation.

- $Y \not\perp Z_C^g | Z_S, Z_F^U$: $Y$ is not conditionally independent of $Z_C^g$ given $Z_S$ and $Z_F^U$.

*Proof.* We first prove that $Z_S$ is conditionally independent of both $Z_C^g$ and $Z_F^U$ given $Y$. According to the d-separation criterion, $Z_S$ and $Z_C^g$, $Z_F^U$ share no common parent variables, and the only path from $Z_C^g$, $Z_F^U$ to $Z_S$ is blocked by $Y$, leading to the conditionally independence between $Z_S$ and $Z_C^g$, $Z_F^U$.

To prove the second statement, we observe the conditional independence $Y \perp [U, E] \mid Z_C^g$. Since $Z_C^g$ is the only cause variable of the target $Y$, it blocks the conditional correlation between $U$, $E$ and $Y$. Therefore, the conditional independence holds.

Finally, we turn to the third statement. We argue that $Y$ is not conditionally independent of $Z_C^g$ given $Z_S$ and $Z_F^U$, as there exists a direct causal path from $Z_C^g$ to $Y$ that cannot be blocked by $Z_S$ and $Z_F^U$. This implies that $Z_F^U$ and $Z_C^g$ play different roles in the prediction process and also justifies this non-independence. $\square$

## B.3. The proof of Proposition 4.1

**Proposition 4.1.** Causal variables $Z_C^g$ should satisfy the following two properties:

- **Invariance under causal intervention**: Based on the first claim in Lemma 3.4, we have $Z_C^g \perp E \mid Y$. Thus, performing an intervention upon $E$ does not make changes to $Z_C^g$.

- **Sufficiency of semantic information**: The causal variable $Z_C^g$ should be causally sufficient to the label prediction, i.e. maximizing $I(Z_C^g; Y)$.

*Proof.* To demonstrate *invariance under causal intervention* property, we first examine the causal relationships among the variables involved. There is no direct or indirect casual path from $E$ to $Z_C^g$ that is not blocked by $Y$. Specifically, the generation of $Z_C^g$ is mainly through $Z_F^U$ which has no direct causal link from $E$. Therefore, $Z_C^g$ and $E$ are d-separated conditioned on $Y$. This means that intervening on $E$ does not affect $Z_C^g$, thus satisfying the invariance property under causal intervention.

To demonstrate the *sufficiency of semantic information* property, we refer to the SCM model in Assumption 3.3, where $Y = \omega(Z_C^g)$. This implies that $Y$ is determined entirely by $Z_C^g$. From the perspective of information, the mutual information is given by $I(Z_C^g; Y) = H(Y) - H(Y \mid Z_C^g)$. Thus, maximizing $I(Z_C^g; Y)$ is to minimize $(Y \mid Z_C^g)$. According to the causal relationships among variables, $Z_C^g$ is the only cause of the target $Y$ and there is no other variable that can provide additional information for predicting $Y$ other than $Z_C^g$, which should minimize $(Y \mid Z_C^g)$, thus satisfying the sufficiency of semantic information property. $\square$

## B.4. The proof of Lemma 4.2 and Lemma 4.3

**Lemma 4.2.** $H(x'|x)$ can be approximated by the conditional probability $p(x'|x)$ as $\mathcal{L}_1$-Laplacians, as follows:

$$H(x'|x) = -\mathbb{E}_{x'}(\log P(x'|x)) \approx \mathbb{E}_x[\|x - \psi(x)\|_1].$$

Similarly, $H(x')$ can be approximated by the marginal distribution of $x'$:

$$H(x') = -\mathbb{E}_{x'}(\log P(x')) \approx \mathbb{E}_x[\|\psi(x)\|_1].$$

*Proof.* The $\mathcal{L}_1$-Laplace distribution is often used to approximate conditional entropy, which effectively captures deviations under noise-perturbed conditions (Carlini et al., 2019; Guo et al., 2023a). In the first approximation, the conditional probability is modeled as a $\mathcal{L}_1$-Laplace distribution centered at $\psi(x)$ with identity covariance,

$$P(x'|x) = \mathcal{L}(x'; \mu(x), I) \propto \exp(-\|x - \psi(x)\|_1). \tag{18}$$

By using this approximation above, the conclusion in Lemma 4.2 holds. Similarly, in the subsequent approximation, we model the distribution using an $\mathcal{L}_1$-Laplace distribution centered at zero,

$$P(x') = \mathcal{L}(x'; \mathbf{0}, I) \propto \exp(-\|\psi(x))\|_1). \tag{19}$$

$\square$

**Lemma 4.3.** Maximizing the diversity loss is approximately equivalent to maximizing the lower bounds of $L_{DIV}$, which satisfy the following inequality:

$$\max_{\psi} \mathcal{L}_{DIV} \geq \max_{\psi} \mathbb{E}_x\left(\left|1 - \frac{\|x\|_1}{\|x'\|_1}\right|\right).$$

It can be observed that generating the counterfactual image $x'$ essentially involves maximizing $\|\|x'\|_1 - \|x\|_1\|_p$.

*Proof.* Since $\|x\|_1$ and $\|x'\|_1$ are positive, the left-hand side of the above inequality can be rewritten as follows:

$$\left|1 - \frac{\|x\|_1}{\|x'\|_1}\right| = \frac{|\|x'\|_1 - \|x\|_1|}{\|x'\|_1}. \tag{20}$$

Notice that for most practical distributions, especially when $\psi(x)$ (or $x'$) is close to $x$, $\|x' - x\|_1$ tends to be greater than or equal to $|\|x'\|_1 - \|x\|_1|$, cause changes in the $L_1$-norm are generally larger compared to element-wise differences. Therefore, it's reasonable to conclude:

$$\mathbb{E}_x\left(\frac{|\psi(x) - x|_1}{|\psi(x)|_1}\right) \geq \mathbb{E}_x\left(\left|1 - \frac{\|x\|_1}{\|x'\|_1}\right|\right). \tag{21}$$

Combining the analysis above, we can derive:

$$\max_{\psi} \mathcal{L}_{DIV} \geq \max_{\psi} \mathbb{E}_x\left(\left|1 - \frac{\|x\|_1}{\|x'\|_1}\right|\right). \tag{22}$$

$\square$

### B.5. The proof of Lemma 4.4

**Lemma 4.4.** The CMI admits the following decompositions:

$$I[Y; Z_F^U \mid \phi^*(X)/Z_F^U] = -H(Y|\phi^*(X)) + H(Y|\phi^*(X)/Z_F^U),$$
$$I[Y; \phi^*(X)/Z_F^U \mid Z_F^U] = -H(Y|\phi^*(X)) + H(Y|Z_F^U).$$

*Proof.* Under our Restructured SCM in Assumption 3.3, our goal is to distinguish $Z_F^U$ from $Z_C^g$ to effectively adapt to test data distributions. It can be observed that we can distinguish them based on their differing behaviors when conditioned on each other in predicting $Y$. In line with (Chen et al., 2024), we adopt CMI as a principled metric to analyze the dependencies among different feature variables. However, (Chen et al., 2024) focuses on extracting causal features and computing the mutual information between causal features and label $Y$, whereas our focus is on eliminating *inactive* causal features $Z_F^U$ from global causal features $Z_C^g$.

The CMI between $X$ and $Y$ given $Z$ can be decomposed into entropy terms as follows:

$$I[X; Y|Z] = H(X|Z) + H(Y|Z) - H(X, Y|Z). \tag{23}$$

Note that when $\phi(X)$ is well trained, the models encourage $\phi^*(X) = Z_F^U \cup Z_C^U$, therefore, $Z_C^U = \phi^*(X)/Z_F^U$.

$$I[Y; Z_F^U|Z_C^U] = H(Y|Z_C^U) + H(Z_F^U|Z_C^U) - H(Y; Z_F^U|Z_C^U). \tag{24}$$

By the definition of conditional entropy (Shannon, 1948), we obtain:

$$H(Z_F^U|Z_C^U) = H(Z_F^U, Z_C^U) - H(Z_C^U), \quad H(Y|Z_F^U, Z_C^U) = H(Y, Z_F^U, Z_C^U) - H(Z_F^U, Z_C^U), \tag{25}$$

where $H(Z_F^U, Z_C^U)$ is the joint entropy of $Z_F^U$ and $Z_C^U$, $H(Z_C^U)$ is the entropy of $Z_C^U$, and $H(Y, Z_F^U, Z_C^U)$ is the joint entropy of $Z_F^U, Z_C^U$ and $Y$. From the above decomposition in Eq. (23), the following equation can be derived:

$$\begin{aligned}
H(Z_F^U|Z_C^U) &= [H(Z_F^U, Z_C^U) - H(Z_C^U)] - [H(Y, Z_F^U, Z_C^U) - H(Z_F^U, Z_C^U)] \\
&= 2H(Z_F^U, Z_C^U) - H(Z_C^U) - H(Y, Z_F^U, Z_C^U) \\
&= 2H(Z_F^U, Z_C^U) - H(Z_C^U) - [H(Y|Z_F^U, Z_C^U) + H(Z_F^U, Z_C^U)] \\
&= H(Z_F^U, Z_C^U) - H(Z_C^U) - H(Y|Z_F^U, Z_C^U) \\
&= H(Z_F^U|Z_C^U) - H(Y|Z_F^U, Z_C^U).
\end{aligned} \tag{26}$$

Then, substituting the above expression into Eq. (24) and simplifying yields:

$$\begin{aligned}
I[Y; Z_F^U|Z_C^U] &= H(Y|Z_C^U) + [H(Z_F^U|Z_C^U) - H(Y|Z_F^U, Z_C^U)] - H(Z_F^U|Z_C^U) \\
&= -H(Y|\phi^*(X)) + H(Y|\phi^*(X)/Z_F^U).
\end{aligned} \tag{27}$$

In the same way, we can prove the second equation in the lemma:

$$I[Y; \phi^*(X)/Z_F^U \mid Z_F^U] = -H(Y|\phi^*(X)) + H(Y|Z_F^U). \tag{28}$$

$\square$

## C. Theoretical analysis of generalization boundary

To analyze the generalization boundary of FedUni, we follow (Mohri, 2018) and use Vapnik–Chervonenkis (VC) dimension theory (Reichenbach, 1971). VC dimension is an important concept in statistical learning theory used to measure the complexity and expressiveness of a hypothesis class, which is closely related to a model's generalization ability. Generally, models with a higher VC dimension indicate that the hypothesis class can represent more complex functions and may perform well on training data but might overfit when applied to testing data.

As mentioned in Section 3, each client in our model consists of a feature extractor and a classifier, and can be formulated as $f_{\theta_c}(\cdot) = w_c(\Phi_c(\cdot))$, where $\theta_c = (w_c, \phi_c)$. Considering only the stages following feature extraction, the global optimization objective is given in Eq. (17). It is noteworthy that all components of the optimization objective are upper-bounded. Without loss of generality, we assume the loss function is bounded above by 1. Let $\mathcal{H}$ be a family of functions that take values of the one-hot vectors of length $k$ for a $k$-classification problem, where the VC-dimension is $d$. Then, for any $\delta > 0$, with probability at least $1 - \delta$, for any $h_{x_i} \in \mathcal{H}$, the following holds:

$$\mathbb{E}[R(h_{x_i})] \leq \mathbb{E}[\hat{R}_S(h_{x_i})] + 2\sqrt{\frac{2d \log \frac{em}{d}}{m}} + \sqrt{\frac{\log \frac{1}{\delta}}{2m}}. \tag{29}$$

where $R(h_{x_i})$ and $\hat{R}_S(h_{x_i})$ are the generalization error and empirical error on sample $S$ respectively. Since the empirical error in our method is upper-bounded, the generalization error has a upper bound as well. To establish this result, we

build upon the analysis in (Mohri, 2018), generalizing it from the centralized binary classification setting to a multi-client, multi-class scenario. Firstly, let $\mathcal{F}$ be a family of functions mapping from $\mathcal{Z}$ to $[0,1]$, in which $\mathcal{Z} = \mathcal{X} \times \mathcal{Y}$ is the input space. Then, for any $\delta > 0$, with probability at least $1 - \delta$ over i.i.d. sample $S$ of size $m$, the following holds for each $f \in \mathcal{F}$:

$$\mathbb{E}[f(z)] \leq \frac{1}{m} \sum_{i=1}^{m} f(z_i) + 2\mathfrak{R}_m(\mathcal{F}) + \sqrt{\frac{\log \frac{1}{\delta}}{2m}}, \tag{30}$$

where $\mathfrak{R}$ is the Rademacher complexity (Shalev-Shwartz & Ben-David, 2014). Our focus now shifts to proving that the right side of Eq. (30) is bounded, with the key challenge lying in demonstrating that $\mathfrak{R}_m(\mathcal{F})$ is bounded.

Let $\mathcal{H}$ be a family of functions that take values in the one-hot vectors of length $k$ for a $k$-classification problem, and $\mathcal{F}$ be the family of loss functions associated with $\mathcal{H}$ for the zero-one loss, i.e., $\mathcal{F} = \{(x, y) \to 1_{h(x) \neq y} : h \in \mathcal{H}\}$. For any sample $S = \{(x_1, y_1), \ldots, (x_m, y_m)\}$ of elements in $(\mathcal{X}, \mathcal{Y})$, let $S_X$ denote its projection onto $\mathcal{X}$. Then, the following relation holds between the empirical Rademacher complexities of $\mathcal{F}$ and $\mathcal{H}$:

$$\mathfrak{R}_S(\mathcal{F}) = \mathfrak{R}_{S_X}(\mathcal{H}). \tag{31}$$

Using Eq. (30) and Eq. (31), we have

$$R(h) \leq \hat{R}_S(h) + 2\mathfrak{R}_m(\mathcal{H}) + \sqrt{\frac{\log \frac{1}{\delta}}{2m}}, \tag{32}$$

where $R(h)$ is the generalization error and $\hat{R}_S(h)$ is the empirical error. According to Massart's Lemma (Massart, 2000) and Sauer's Lemma (Sauer, 1972), which together establish an explicit relationship among the Rademacher complexity, the growth function, and the VC-dimension, we can derive the following result based on Eq. (32): for any $\delta > 0$, with probability at least $1 - \delta$, for every function $h \in \mathcal{H}$, the following upper bound on the generalization error $R(h)$ holds:

$$R(h) \leq \hat{R}_S(h) + 2\sqrt{\frac{2 \log \Pi_{\mathcal{H}}(m)}{m}} + \sqrt{\frac{\log \frac{1}{\delta}}{2m}} \leq \hat{R}_S(h) + 2\sqrt{\frac{2d \log \frac{em}{d}}{m}} + \sqrt{\frac{\log \frac{1}{\delta}}{2m}}, \tag{33}$$

where $\Pi_{\mathcal{H}}(m)$ is the growth function of $\mathcal{H}$ and $d$ is the VC-dimension of $\mathcal{H}$. Since Eq. (33) holds for all clients in our method, we now focus on the expected generalization error for each client. In the multi-client scenario, the result is shown in the following equation, thereby proving the theoretical generalization bound of FedUni:

$$\mathbb{E}[R(h_{x_i})] \leq \mathbb{E}[\hat{R}_S(h_{x_i})] + 2\sqrt{\frac{2d \log \frac{em}{d}}{m}} + \sqrt{\frac{\log \frac{1}{\delta}}{2m}}. \tag{34}$$

# D. More Details about Experiments

### D.1. Description of the dataset

**Waterbirds** is a synthetic dataset that combines bird photographs from the Caltech-UCSD Birds-200-2011 (CUB) dataset (Welinder et al., 2010) with backgrounds from the Places dataset (Zhou et al., 2014). This dataset provides a binary classification task for 'landbird' and 'waterbird', and the *background* is spuriously correlated with the class. We use $p_s$ to quantify the degree of spurious correlation, where landbirds have a probability of $p_s$ with a land background and $(1 - p_s)$ with a water background.

**Colored-MNIST (CMNIST)** is a variant of the original MNIST (LeCun et al., 1998) dataset via rearranging the images of digits 0-4 into one class (class 0) and the images of digits 5-9 into another class (class 1). The *color* is spuriously correlated with the class, where each digit with label 0 is colored green/red with probability $p_s/1 - p_s$, and this probability is reversed for label 1.

**Colored-Fashion MNIST (CFMNIST)** is constructed from Fashion-MNIST (Xiao et al., 2017) using the same strategy as CMNIST.

**Digits** consists of 5 different datasets: MNIST (LeCun et al., 1998), SVHN (Netzer et al., 2011), MNISTM (Ganin & Lempitsky, 2015), SYN (Ganin & Lempitsky, 2015) and USPS (Denker et al., 1988). Each dataset is considered a unique domain which may be different from the rest of the domains in font style, background, and color.

**PACS** consists of a total of 9,991 samples, each with a dimension of (3, 224, 224). It includes 7 classes and covers four domains: *art, cartoon, photo, sketch*.

### D.2. Implementation Details

For Waterbirds, CMNIST, and CFMNIST datasets, the degree of spurious correlation $p_s$ for each source client is randomly sampled from $[0.85 : 1.0]$, while for the test data $p_s$ varies from $0.$ to $1.0$. Considering the heterogeneity across clients, the training data on each client are sampled from a subset of the original classes. Specifically, we conduct experiments on 10 clients. We distribute 10 waterbird species (4 separated and 6 overlapped) and 19 landbird species (15 separated and 4 overlapped) to each client on the Waterbirds dataset. And on the CMNIST/CFMNIST dataset, we re-label the data with labels $\{0, 1, 2, 3, 4\}$ as class 0, and the data with labels $\{5, 6, 7, 8, 9\}$ as class 1. Furthermore, we randomly select two different digit subclasses from each of class 0 and class 1. Since the test distribution in the real world is unknown, the model may be deployed to source clients, which share the same distribution as the training data, or to unseen clients, whose data distribution shifts from the training process. We aim to enhance OOD generalization performance while maintaining the discriminative ability on the ID data distribution as much as possible. We refer to the results for $p_s \in \{1.0, 0.9, 0.8\}$ as the in-distribution (ID) case, since this degree is close to the training data distribution, while the remaining values correspond to the OOD case.

For Digits and PACS datasets, we follow the *'leave-one-out'* rule, where we choose one domain as target domain, train the model on all remaining domains, and evaluate on the target domain. We conduct experiments on 10 clients, the data on each client are sampled from the same domain, and the domains on different clients can be repeated. For example, in the Digits dataset, there are 5 domains in total, with training clients covering 4 of these domains. The number of clients in each training domain is 2, 2, 2, and 4, respectively, and each domain contains 1000 training samples. In the PACS dataset, there are 4 domains in total, with training clients covering 3 of these domains. The number of clients in each training domain is 3, 3, and 4, respectively, and each domain contains 750 training samples.

## E. Additional Method Details

### E.1. Structure of the counterfactual generator

Our generator $\psi(x; \theta_\psi)$ aims to synthesize novel and diverse samples to extend the distribution of the source domain. We chose a model architecture that is widely used in the field of style transfer (Guo et al., 2023a; Yang et al., 2024), due to its proven effectiveness in capturing and transferring domain information. $\psi$ consists of two convolution layers and $AdaIN_{noise}$. The output synthesized by $\psi(x; \theta_\psi)$ is defined as follows:

$$\psi(x; \theta_\psi) = Conv(AdaIN_{noise}(Conv(x))), \tag{35}$$

where $Conv$ represents the convolution layers and $AdaIN_{noise}$ is a variant of adaptive instance normalization (Huang & Belongie, 2017). To achieve counterfactual generation, $AdaIN_{noise}$ replaces affine transformation parameters with style mean $\hat{\mu} \in \mathbb{R}^c$ and style variance $\hat{\sigma} \in \mathbb{R}^c$, while simultaneously incorporating the variable noise $\hat{\eta} \in \mathbb{R}^{h \times w \times c}$ into the semantic content, with $h$, $w$ and $c$ indicating the height, width, and channel, respectively.

## F. Additional Experimental Results.

### F.1. Model attention visualization.

To demonstrate that our method relies on causal features rather than non-causal ones, we generate visual explanations using Grad-CAM (Selvaraju et al., 2017). The models are trained on the Waterbirds dataset using various FL methods. Grad-CAM, an extension of CAM, is a tool that provides visual insights into CNN-based image classifiers by highlighting image regions that contribute most to the model's predictions, thus allowing us to assess the features the model uses as the basis for its classification decisions.

The detailed results can be found in Fig. 8, which shows that the model trained with FedUni focuses more on the causal feature regions relevant to classification, such as the body parts and feather textures of birds. In contrast, the comparative

models rely on spurious features, such as the background. These findings provide strong evidence that our method can more effectively extract invariant causal features while avoiding spurious associations.

### F.2. Mitigation of spurious correlations.

We further evaluate the extent of spurious associations by training models using different FL methods on three datasets: CMNIST, CFMNIST, and Waterbirds. The degree to which each model relies on spurious features, such as background and color, is quantified by the probability $p_s$.

In Table 4, 5, and 6, we demonstrate the relationship between test accuracy and $p_s$ for each model. Our proposed model outperforms the SOTA methods by exhibiting lower dependence on spurious associations across all three datasets. This indicates that our model effectively mitigates reliance on spurious associations.

| p = | 1 | 0.9 | 0.8 | 0.7 | 0.6 | 0.5 | 0.4 | 0.3 | 0.2 | 0.1 | 0 |
|---|---|---|---|---|---|---|---|---|---|---|---|
| FedAvg | 87.92 | 86.3 | 85.07 | 82.84 | 82.26 | 80.35 | 78.99 | 77.40 | 75.59 | 74.26 | 70.18 |
| -worst | 85.01 | 82.67 | 82.66 | 78.81 | 79.18 | 77.44 | 75.71 | 72.86 | 71.79 | 70.18 | 67.5 |
| FedProx | 88.2 | 86.41 | 85.55 | 82.91 | 81.8 | 80.36 | 78.06 | 76.69 | 75.02 | 73.69 | 71.88 |
| -worst | 84.3 | 83.07 | 80.78 | 79.6 | 77.43 | 76.77 | 74.43 | 72.32 | 70.72 | 69.49 | 67.86 |
| Scaffold | 87.89 | 86.2 | 84.97 | 82.49 | 82.26 | 80.19 | 78.85 | 77.18 | 75.79 | 74.02 | 72.54 |
| -worst | 84.33 | 83.25 | 81.83 | 79.78 | 78.66 | 78.16 | 76.19 | 74.21 | 73.02 | 70.22 | 68.95 |
| Moon | 86.15 | 84.69 | 83.93 | 82.53 | 81.64 | 80.56 | 78.9 | 77.99 | 77.01 | 75.67 | 74.64 |
| -worst | 80.78 | 79.37 | 79.89 | 78.84 | 76.54 | 76.42 | 75.31 | 73.93 | 72.84 | 72.13 | 70.89 |
| FedSR | 87.99 | 86.55 | 85.49 | 83.45 | 82.52 | 81.2 | 79.41 | 78.28 | 77.06 | 75.74 | 74.26 |
| -worst | 83.25 | 82.89 | 81.31 | 80.69 | 77.62 | 77.08 | 74.91 | 74.64 | 73.57 | 72.92 | 68.93 |
| FedIIR | 88.1 | 86.14 | 85.61 | 82.7 | 81.88 | 80.27 | 78.79 | 76.54 | 75.25 | 73.52 | 72.01 |
| -worst | 85.01 | 83.03 | 82.19 | 79.54 | 78.16 | 77.44 | 74.82 | 72.32 | 71.66 | 70.04 | 68.39 |
| FedDG-GA | 87.35 | 85.64 | 84.79 | 82.86 | 82.38 | 79.95 | 78.63 | 77.31 | 75.84 | 74.49 | 73.13 |
| -worst | 82.54 | 82.36 | 81.13 | 80.07 | 79.4 | 76.17 | 74.19 | 73.11 | 70.94 | 69.49 | 67.51 |
| FedSDR | 87.64 | 86.22 | 85.74 | 84 | 83.24 | 81.86 | 80.78 | 79.4 | 78.69 | 77.7 | 76.67 |
| -worst | 84.52 | 82.46 | 82.54 | 81.72 | 79.72 | 78.67 | 77.86 | 76.43 | 75.54 | 73.65 | 71.84 |
| Ours | 89.45 | 88.38 | 87.81 | 86.33 | 85.64 | 84.25 | 83.08 | 82.24 | 81.4 | 80.61 | 79.41 |
| -worst | 86.07 | 85.26 | 84.3 | 84.13 | 83.25 | 81.39 | 80.78 | 79.11 | 78.21 | 78.16 | 76.61 |

*Table 4.* The relationship between average and worst test accuracy and test distribution specified by $p_s$ on **Waterbirds** dataset. (Rows without background color show the average test accuracy results.)

### F.3. Convergence behavior.

To provide a clear illustration of the convergence behavior of our method, we plot the test accuracy over global communication rounds on the Digits dataset, as shown in Fig. 9. In the early stages of training, the test accuracy shows some fluctuations because the model has not yet fully captured the causal features. However, as the number of communication rounds increases, the model gradually learns more robust feature representations, enhancing its ability to handle diverse data distributions. Eventually, the curve becomes stable. This trend demonstrates that our method steadily converges throughout the training process and maintains stable performance in later stages, indicating its effectiveness and robustness in heterogeneous data scenarios.

## G. Discussion of Limitations and Future Direction.

In this section, we discuss the limitations of our method and the potential solutions.

Despite the promising results our proposed method has achieved in reducing the reliance on spurious associations, it still suffers from some limitations.

Firstly, in terms of computation, compared with the FedAvg method, the CMI filter and counterfactual generator included in our proposed method bring a relatively large computational burden. This restricts the scale of the model and the amount

| p = | 1 | 0.9 | 0.8 | 0.7 | 0.6 | 0.5 | 0.4 | 0.3 | 0.2 | 0.1 | 0 |
|---|---|---|---|---|---|---|---|---|---|---|---|
| FedAvg | 96.72 | 91.92 | 87.1 | 82.46 | 77.67 | 72.73 | 68.53 | 63.28 | 59.21 | 53.92 | 49.18 |
| -worst | 90.5 | 85.4 | 82.3 | 74.9 | 67 | 61 | 54.6 | 48.2 | 39.7 | 32.7 | 26.1 |
| FedProx | 96.4 | 91.81 | 87.31 | 82.28 | 77.7 | 73.05 | 68.81 | 63.6 | 59.55 | 54.56 | 49.97 |
| -worst | 90.3 | 85.5 | 81.2 | 75.9 | 68.2 | 63 | 56.1 | 49 | 40.8 | 34.2 | 27.6 |
| Scaffold | 95.8 | 91.32 | 86.81 | 81.92 | 77.58 | 72.83 | 68.68 | 63.64 | 59.71 | 54.77 | 50.27 |
| -worst | 88.7 | 84.5 | 80.9 | 75.9 | 68.8 | 62.4 | 56.8 | 48.8 | 42.4 | 35.6 | 29.1 |
| Moon | 94.31 | 92.89 | 91.61 | 90.54 | 88.93 | 88.04 | 86.49 | 85.41 | 84.36 | 82.86 | 81.53 |
| -worst | 72.3 | 69.2 | 64.5 | 61.6 | 58.7 | 55 | 50.4 | 50.5 | 46.8 | 42.8 | 39.7 |
| FedSR | 97.18 | 93.2 | 89.39 | 84.74 | 80.96 | 76.46 | 72.52 | 68.69 | 64.97 | 60.18 | 56.34 |
| -worst | 90.3 | 85.7 | 82.3 | 77.7 | 72.45 | 66.7 | 61.9 | 56.2 | 49 | 41.8 | 37.4 |
| FedIIR | 96.22 | 91.77 | 87.25 | 82.4 | 77.93 | 73.39 | 69.3 | 64.14 | 60.38 | 55.33 | 50.79 |
| -worst | 89.3 | 84.9 | 80.2 | 74.9 | 67.5 | 62.5 | 55.4 | 49.4 | 41.5 | 35.1 | 28.6 |
| FedDG-GA | 96.92 | 92.44 | 87.53 | 82.68 | 78.2 | 73.38 | 69.4 | 64.1 | 60.16 | 55.22 | 50.55 |
| -worst | 92.7 | 88.3 | 81.8 | 76 | 69.2 | 62 | 55.7 | 48 | 42 | 34.2 | 26.9 |
| FedSDR | 96.73 | 92.15 | 87.5 | 82.47 | 78.07 | 73.32 | 68.83 | 63.77 | 59.77 | 54.71 | 50.01 |
| -worst | 88.9 | 83.7 | 80.5 | 75.3 | 69.6 | 61.2 | 56.7 | 48.2 | 42.7 | 35.1 | 27.8 |
| Ours | 96.57 | 95.16 | 93.8 | 92.37 | 90.92 | 89.46 | 87.88 | 86.46 | 85.39 | 83.75 | 82.28 |
| -worst | 92 | 89.5 | 86.1 | 82.1 | 78 | 74.7 | 69.7 | 68.4 | 64.2 | 60 | 56.4 |

*Table 5.* The relationship between average and worst test accuracy and test distribution specified by $p_s$ on **CMNIST** dataset. (Rows without background color show the average test accuracy results.)

| p = | 1 | 0.9 | 0.8 | 0.7 | 0.6 | 0.5 | 0.4 | 0.3 | 0.2 | 0.1 | 0 |
|---|---|---|---|---|---|---|---|---|---|---|---|
| FedAvg | 93.13 | 91.96 | 91.14 | 90.4 | 89.31 | 88.38 | 87.21 | 86.61 | 85.79 | 84.72 | 83.71 |
| -worst | 66.7 | 64.4 | 61.2 | 59.9 | 58.4 | 55 | 52 | 52.2 | 48.8 | 46.9 | 44.5 |
| FedProx | 95.23 | 93.62 | 92.08 | 91.1 | 89.03 | 88.07 | 86.28 | 84.92 | 83.57 | 82.05 | 80.54 |
| -worst | 78.3 | 74.4 | 68.1 | 66 | 60.4 | 56.8 | 50.4 | 50 | 45.6 | 40.7 | 36.8 |
| Scaffold | 94.73 | 93.29 | 91.93 | 90.81 | 89.16 | 88.12 | 86.56 | 85.35 | 84.29 | 82.73 | 81.36 |
| -worst | 74.4 | 71.4 | 66.3 | 63.2 | 60.2 | 56.5 | 52.2 | 51.9 | 48.2 | 44.3 | 41.1 |
| Moon | 95.53 | 92.01 | 88.38 | 84.62 | 80.82 | 77.44 | 74.01 | 70.16 | 67.18 | 63.19 | 59.5 |
| -worst | 86.1 | 83.1 | 80.2 | 77 | 72.5 | 67.3 | 61.7 | 55.8 | 51.3 | 45.4 | 39.6 |
| FedSR | 93.4 | 92.34 | 91.14 | 90.19 | 88.62 | 87.89 | 86.18 | 85.68 | 84.48 | 83.27 | 81.99 |
| -worst | 69.5 | 67.2 | 62.7 | 61.2 | 58.5 | 55.9 | 51.2 | 50.6 | 48.4 | 45.9 | 43 |
| FedIIR | 93.53 | 92.3 | 91.26 | 90.43 | 89.14 | 88.22 | 86.86 | 86.04 | 85.07 | 83.96 | 82.78 |
| -worst | 68.3 | 65.7 | 62.1 | 60.3 | 58.6 | 55.1 | 51.7 | 51.8 | 48.2 | 45.6 | 43.2 |
| FedDG-GA | 93.2 | 91.95 | 91.07 | 90.31 | 89.19 | 88.13 | 86.97 | 86.32 | 85.41 | 84.29 | 83.26 |
| -worst | 66.8 | 64.3 | 61.1 | 59.7 | 58.3 | 54.6 | 51.5 | 52 | 48.5 | 46.4 | 44.1 |
| FedSDR | 95.34 | 93.71 | 92.13 | 90.95 | 89.04 | 87.82 | 86.21 | 84.68 | 83.32 | 81.77 | 80.14 |
| -worst | 77.5 | 74.1 | 68.9 | 65.7 | 61.5 | 57.8 | 53.6 | 51.6 | 47.8 | 44.4 | 40.5 |
| Ours | 94.95 | 93.91 | 92.83 | 91.9 | 90.61 | 89.79 | 88.73 | 87.68 | 86.67 | 85.58 | 84.51 |
| -worst | 76.5 | 74.5 | 69.4 | 66.6 | 63.5 | 60.7 | 57.2 | 55.6 | 52.5 | 49.3 | 46.2 |

*Table 6.* The relationship between average and worst test accuracy and test distribution specified by $p_s$ on **CFMNIST** dataset. (Rows without background color show the average test accuracy results.)

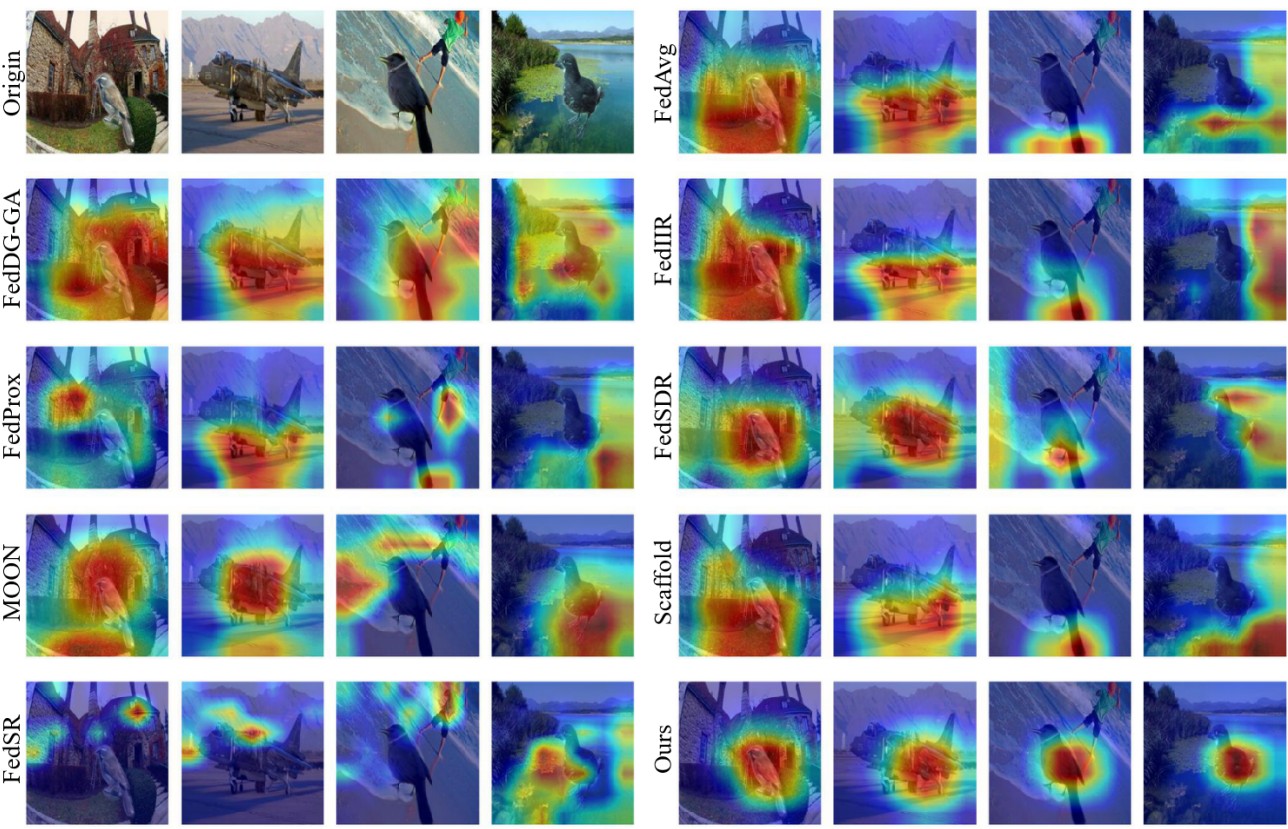

*Figure 8.* Visualization of the effects of some federated learning methods (including ours) on the Waterbirds dataset. The red areas are important for the prediction of the correct class, and the blue areas are vice versa.

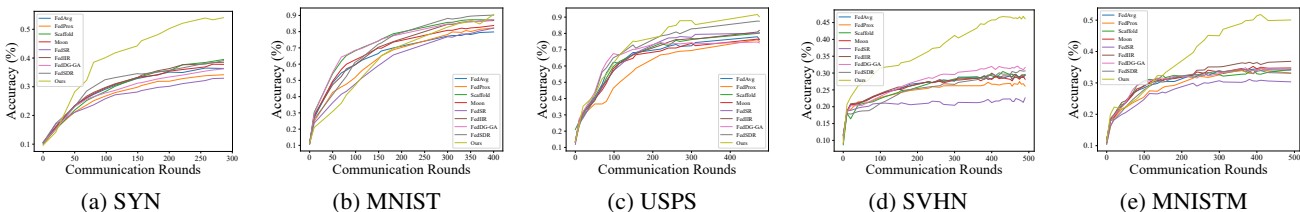

(a) SYN      (b) MNIST      (c) USPS      (d) SVHN      (e) MNISTM

*Figure 9.* Accuracy curves with global communication rounds increase.

of data that can be processed. As a result, our method fits better in cross-silo FL scenarios where devices have sufficient computing power.

Secondly, device heterogeneity has not been considered. In FL, different clients may have diverse hardware conditions, including network environments and computing capabilities. Some clients can handle complex causal feature extraction and selection, while others may struggle to complete these tasks due to limitations in computing capabilities, network, or memory. On the other hand, the datasets on different clients may also vary. Some clients may have very few or low-quality datasets, which can affect the local learning effect and, in turn, the overall performance. It is important to note that the current framework proposed in this paper assumes uniform computational resources across all clients. This assumption simplifies the experimental setup but does not fully capture the complexity of real-world scenarios involving heterogeneous devices and resource constraints.

To address the aforementioned limitations, several promising research directions can be explored. A key area is optimizing the computational efficiency of our method. By exploring more efficient algorithms and techniques, we can reduce the computational burden and make the method applicable to a wider range of model scales and data amounts. Additionally, designing customized lightweight approximation schemes for clients with low computing power and unstable network conditions could further enhance the adaptability of our method.

