# OpenReview forum: "Causality Inspired Federated Learning for OOD Generalization"
_ICML.cc/2025/Conference — ICML 2025 poster_

### Official Review · Reviewer_MJM2 · 2025-03-07

**Overall Recommendation:** 3

**Summary:**

I am not familiar with the field of this submission. I suggest the AC find another reviewer and disregard my comments. Apologies for the inconvenience.

**Claims And Evidence:**

I am not familiar with the field of this submission. I suggest the AC find another reviewer and disregard my comments. Apologies for the inconvenience.

**Essential References Not Discussed:**

I am not familiar with the field of this submission. I suggest the AC find another reviewer and disregard my comments. Apologies for the inconvenience.

**Experimental Designs Or Analyses:**

I am not familiar with the field of this submission. I suggest the AC find another reviewer and disregard my comments. Apologies for the inconvenience.

**Methods And Evaluation Criteria:**

I am not familiar with the field of this submission. I suggest the AC find another reviewer and disregard my comments. Apologies for the inconvenience.

**Other Comments Or Suggestions:**

No

**Other Strengths And Weaknesses:**

No

**Questions For Authors:**

No

**Relation To Broader Scientific Literature:**

I am not familiar with the field of this submission. I suggest the AC find another reviewer and disregard my comments. Apologies for the inconvenience.

**Theoretical Claims:**

I am not familiar with the field of this submission. I suggest the AC find another reviewer and disregard my comments. Apologies for the inconvenience.

---

> ### Author Rebuttal · Authors · 2025-04-01
>
> Thank you for your rigorous and responsible review.

---

### Official Review · Reviewer_Lsn9 · 2025-03-09

**Overall Recommendation:** 3

**Summary:**

The authors identify a limitation in federated learning, arguing that existing methods primarily capture invariant causal features across clients/environments while failing to account for variant causal features. To address this, they propose a method designed to capture both invariant and variant causal features—both of which are direct causes of the target variable, y. Their proposed architecture, FedUni, is trained to extract all possible causal features from input data, acknowledging that some extracted features may be non-causal. To mitigate the risk of incorporating spurious correlations, they introduce a feature compressor. Additionally, they incorporate a causal intervention module on the client side, leveraging a counterfactual generator to create counterfactual examples.

The authors provide extensive experiments and theoretical analysis, claiming that their method significantly enhances out-of-distribution (OOD) generalization.

**Claims And Evidence:**

While the approach appears well-motivated, the effectiveness of capturing variant causal features—especially under real-world federated settings—remains an open question. A deeper examination of the robustness of their counterfactual generation and feature compression mechanisms would be valuable in assessing the practical impact of this work.

**Essential References Not Discussed:**

Not that I am aware of.

**Experimental Designs Or Analyses:**

Experiments are reasonable.

**Methods And Evaluation Criteria:**

The motivation and proposed method are well-founded, and the choice of benchmark datasets is appropriate.

**Other Comments Or Suggestions:**

Already mentioned in above.

**Other Strengths And Weaknesses:**

The paper has a good motivation. However, it lacks clarity to explain and support each component.

In lines 281–283, the authors state, "Firstly, with the causal intervention module, Z_S can be eliminated by minimizing the causal effect of environmental changes." However, this claim seems questionable, as the authors themselves acknowledge that some detected causal features may, in fact, be non-causal.

It remains unclear how misidentified non-causal features are effectively reduced while true causal features are amplified. Further clarification on this mechanism would be beneficial.

**Questions For Authors:**

Please address my comment in "Other Strengths And Weaknesses".

**Relation To Broader Scientific Literature:**

It is relevant to the broader machine learning community.

**Theoretical Claims:**

No concern.

---

> ### Author Rebuttal · Authors · 2025-04-01
>
> In response to the reviewer's concern that “some detected causal features may, in fact, be non-causal,” we clarify that the features described above are indeed causal. The misunderstanding appears to stem from our use of the misleading term “**fake causal features**” to denote causal features that **exist in general but are absent in the current image**. Importantly, “fake causal features” do not refer to ***misidentified*** non-causal features.  Perhaps a more precise term would be “**inactive causal features**” and we will replace “fake causal features” with “inactive causal features” in the final version.
>
> More specifically, after eliminating the non-causal features $Z_S$ in lines 281–283, the remaining causal features $Z_C^g$ can be further divided into **active causal features** $Z_C^U$ and **inactive causal features** $Z_F^U$. The feature compressor then performs adaptive feature selection based on the test data distribution $U$, retaining the active causal features $Z_C^U$ and discarding the inactive causal features $Z_F^U$ that are not present in the current data. For example, a *cat paw* is a valid causal feature for classifying a cat. However, in an image where only a *cat head* is visible, the *cat paw* becomes an inactive causal feature. Inactive causal features may confuse the model, leading to performance degradation, so they should be excluded from the final features.
>
> Thus, the introduction of the counterfactual generator enables the elimination of non-causal features $Z_S$, while the feature compressor eliminates inactive causal features $Z_F^U$ and amplifies active causal features $Z_C^U$.

---

### Official Review · Reviewer_KHjr · 2025-03-11

**Overall Recommendation:** 3

**Summary:**

This paper addresses the challenge of OOD generalization in FL by rethinking how causal features are extracted across clients. The authors argue that instead of limiting the global feature extractor to invariant features (like the traditional FL methods), it should capture the union of all causal features present in the data, thereby preserving richer and more diverse information that can enhance OOD performance by leveraging both collaborative training and targeted causal interventions. The proposed FedUni comprising three core components. 1) A comprehensive feature extractor is designed to identify all potential causal features from any input across clients. 2) A fake causal feature compressor is employed to filter out client-specific or spurious features that do not contribute to the target task. 3) A causal intervention module on the client side uses a trainable counterfactual generator to create modified examples that simulate distribution shifts while preserving semantic content. Experimental results on different datasets and theoretical analysis are provided in the paper.

**Claims And Evidence:**

The process of capturing causal features while rejecting fake ones relies heavily on the performance of a counterfactual generator. This separation could be in practice challenging and dependent on the model type, architecture, size, hyperparameters, etc. In this paper, a very simplistic model of two convolution layers along with some normalization was considered; but, how we can justify using such simplistic model for the purpose, especially for practical size images (not to small pixel size)?

**Essential References Not Discussed:**

See above comment. Relevant references for PFL for OOD generalization should be cited (and if possible, the performance should be compared). Also, FOOGD: Federated Collaboration for Both Out-of-distribution Generalization and Detection (NeurIPS 2024) should be considered (and if possible, the performance should be compared).

**Experimental Designs Or Analyses:**

1.	The experimental evaluation in this paper is undermined by the use of outdated and overly simplistic models, namely ResNet-18, a one-hidden layer MLP, and AlexNet. These choices do not reflect the current SOTA architectures (ResNet-50/101, ViT, etc) that are standard in top-tier research. These modern architectures have fundamentally different inductive biases, scalability, and computational characteristics. Therefore, the results currently presented in this paper may not reliably generalize to realistic settings.
2.	The number of clients seems to be small, N=10, to capture real-world scenarios. In many recent papers on FL, much more clients are often considered, e.g., 100.
3.	Key practical issues of FL such as non-iid data and client sampling have not been considered.

**Methods And Evaluation Criteria:**

Yes.

**Other Comments Or Suggestions:**

The proposed method can be computationally demanding on the client side due to the extra modules, which raises a question of scalability. In the current experimental setup, the model architecture and number of clients are all small. More experiments on large-scale or highly heterogeneous federated settings would help assess the scalability of the proposed scheme.

**Other Strengths And Weaknesses:**

Another weakness: In the paper, the intervention module is a key component, which generates counterfactual examples that simulate distributions shifts. However, it is not clear how the authors can guarantee that these generated examples accurately simulate distribution shifts without altering the underlying semantics and how sensitive the method is to potential failures of proper generation? In Appendix E, the authors clarify that the generator is a simplistic model composed of two convolution layers and some adaptive normalization. But, I am not convinced that this overly simplistic network can generate high-quality realistic samples that simulate distributions shifts, without changing the underlying semantics.

**Questions For Authors:**

My concerns and questions have been raised in the above comments.

**Relation To Broader Scientific Literature:**

The problem of OOD generalization can be also addressed through personalized FL (PFL). It seems that the authors claim that the PFL approach is limited, by citing a couple of works such as (Tang 2023, 2024). But, the literature of PFL is very rich, and some of them explicitly tackle the issue of OOD generalization while numerous others tackle the issue implicitly (but, possibly effectively). Few examples are:

[Ref 1] Exploiting Personalized Invariance for Better Out-of-distribution Generalization in Federated Learning

[Ref 2] Personalized Federated Learning with Contextualized Generalization

[Ref 3] Ditto: Fair and Robust Personalized Federated Learning

I am hoping to see fair and proper comparison against those approaches of PFL.

**Theoretical Claims:**

Seem to be correct.

---

> ### Author Rebuttal · Authors · 2025-04-01
>
> Thanks for your suggestions, and we will include the citation you pointed out in the final version.
>
> **Q1: Experimental Designs**
>
> We conduct additional experiments in response to the suggestions about experimental designs.
>
> **Add Updated Model Architectures**:  Following the reviewer's suggestion, we include experiments with ResNet-50 and ViT on PACS dataset, and our method still achieves superior performance. Notably, the selected models in original submission was guided by **the experimental settings of existing works** to ensure a fair comparison.
>
> |ResNet50|Art|Cartoon|Photo|Sketch|Avg|
> |-|-|-|-|-|-|
> |FedAvg|75.07|75.47|92.80|76.20|79.88|
> |FedSR|74.27|71.60|92.13|67.87|76.47|
> |FedGD-GA|77.47|72.80|**94.27**|69.73|78.57|
> |FedIIR|74.80|71.87|91.33|71.87|77.47|
> |Ours|**80.27**|**75.87**|93.27|**79.60**|**82.25**|
>
> |ViT-Base-32|Art|Cartoon|Photo|Sketch|Avg|
> |-|-|-|-|-|-|
> |FedAvg|80.01|74.53|93.07|69.73|79.33|
> |FedSR|73.60|68.13|87.33|65.07|73.53|
> |FedGD-GA|80.34|71.27|93.07|65.87|77.63|
> |FedIIR|78.93|72.67|87.87|68.93|77.10|
> |Ours|**80.40**|**74.60**|**94.93**|**78.13**|**82.02**|
>
> **Increase Client Number and Add Client Sampling**: We added experiments on the CMNIST dataset with 100 clients, adopting the widely used random client sampling strategy by selecting 20 clients per round for training.
>
> |Test Acc (%)|FedAvg|FedProx|Scaffold|Moon|FedSR|FedIIR|FedDG-GA|FedSDR|Ours|
> |-|-|-|-|-|-|-|-|-|-|
> |ID|97.65|97.86|98.25|97.66|98.45|96.04|94.5|98.19|**99.61**|
> |OOD|76.48|76.59|83.51|76.79|80.08|72.69|73.86|76.77|**93.87**|
>
> **Consider Non-iid data**: Reviewer suggested considering non-IID data. However, we **have already considered non-IID settings**. The experiment results presented in Tables 1 and 2 on page 7 of the original submission were all obtained under non-IID conditions, addressing both class imbalance and covariate shift. A detailed description of the non-IID setup can be found in Appendix D.2.
>
> **Q2: Broader Scientific Literature and References**
>
> We closely follow the reviewer’s suggestion to conduct comparisons, and we will cite these works in the final version. As we have **cited the advanced versions** from the same team (Tang 2023, 2024), the earlier versions Ref [1,2] were not included in the original submission.
>
> We evaluate the provided methods on the CMNIST dataset with 100 clients. We abbreviate the method in Ref [1] as DRC. Compared to FOOGD, Ditto, and CG-PFL, our method explicitly considers the extraction of causal features. In contrast to DRC, which focuses on preserving causal features specific to individual clients, our approach retains causal features across all clients. The experimental results further demonstrate the effectiveness of our motivation.
>
> |Test Acc (%)|DRC [ref 1]|CG-PFL [ref 2]|Ditto [ref 3]|Ours|
> |-|-|-|-|-|
> |ID|97.17|98.04|96.15|**99.61**|
> |OOD|76.75|79.94|65.23|**93.87**|
>
> | | Test Acc (%)|
> |-|-|
> |FOOGD|79.61|
> |Ours|**90.26**|
>
> **Q3: Question about Counterfactual Generator**
>
> **How semantic information is preserved:** We need to clarify that in the original submission we explicitly addressed this issue during model training by incorporating a loss term $L_{REG}$ , which encourages the preservation of semantic information. In particular, we provide a detailed explanation in Section 4.1(2) *Preserve semantic information* and offer a theoretical proof of its effectiveness in Lemma 4.3 on Page 5.
>
> **Generation Sensitivity:** We need to clarify that in the original submission we have defined the hyperparameter $\alpha$ to measure sensitivity to generation quality, where $\alpha = 0$ indicates generation without semantic constraints, and larger values of $\alpha$ impose stronger constraints. As shown in Fig. 7(a) on page 8 of the original submission, the accuracy only drops from 90.4% to 88.5% when switching from the best setting to a setting without semantic constraints—a gap of only 1.9%. This demonstrates that FedUni is robust with respect to generation quality and hyperparameters.
>
> **Why the simple model architecture works:** Our counterfactual generator does not generate images from noise, instead, it **adds noise (i.e., distribution shifts) to the original image**. Since the original image remains part of the input, a simple model architecture is sufficient to simulate distribution shifts. Furthermore, this architecture is widely used in style transfer methods [1,2,3] and has been shown to efficiently generate distribution shifts while preserving semantic information.
>
>
> [1] Huang, et al. Arbitrary Style Transfer in Real-time with Adaptive Instance Normalization. 2017.
>
> [2] Guo, et al. Single Domain Generalization via Unsupervised Diversity Probe. 2023.
>
> [3] Yang, et al. Practical Single Domain Generalization via Training-time and Test-time Learning. 2024.
>
> **Q4: Computation Cost**
>
> Please refer to our response to Reviewer FyJy 4.  FedUni achieves significantly higher accuracy, while maintaining comparable computational cost to many state-of-art approaches.

---

### Official Review · Reviewer_FyJy · 2025-03-13

**Overall Recommendation:** 4

**Summary:**

The paper proposes FedUni, a framework for federated learning. The framework extracts causal features from different clients in training and flexibly selects the features applicable for the target client. It differs from the existing methods in terms of not using a fixed sharing feature pool. The paper provides experiments to show the improvement brought by the novel design.

**Claims And Evidence:**

The theoretical claims are well-supported by proofs and explanations. I found them easy to follow and understand. The novel specification of SCM looks natural to me, and the advantages are well presented in the experiments.

**Essential References Not Discussed:**

N/A

**Experimental Designs Or Analyses:**

The design (Sec 5.1) and analysis (Sec 5.2) both look solid to me.

**Methods And Evaluation Criteria:**

The proposed method and evaluation makes sense to the specific problem.

**Other Comments Or Suggestions:**

N/A

**Other Strengths And Weaknesses:**

I like the presentation of the paper. Though it is mathematically dense and has a lot of content both theoretically and empirically, it does not cause a burden to read.


It could be helpful to make the text font in some figures (Fig 5, 7) larger.

**Questions For Authors:**

1. What is the computational cost of the current method? I see it is briefly mentioned in Appendix G. I’m wondering if the authors could provide details from either experiments (actual run time) or theories (time complexity/asymptotic convergence rate).

2. How should practitioners know when to choose FedUni over the prior methods?

**Relation To Broader Scientific Literature:**

The design of fake causal features is a smart and novel idea. It may be useful for the federated learning tasks where the target client requires domain knowledge (not just common knowledge) from a specific client in training. Yet for the tasks that the training clients are very similar to each other, the method may not make a large difference despite requiring more computational cost.

**Theoretical Claims:**

I checked Lemma 3.2, 3.4, 4.2. For Lemma 3.2 and 3.4, the techniques are pretty standard and look correct to me.


I think the $\approx$ notion looks hand-wavy in Lemma 4.2, thus I suggest making a more rigorous argument.

---

> ### Author Rebuttal · Authors · 2025-04-01
>
> Thanks for your valuable suggestions, and we will enlarge the text font in Fig 5, 7 in the final version.
>
> **Q1: Computational Cost**
>
> We provided the run time per communication round and a theoretical proof for the convergence rate. Due to space limitations, we provide a brief proof here, and the full proof will be included in the appendix of the final version.
>
> Firstly, we measured the runtime per communication round (20 clients with 5 local training epochs)  on two commonly used GPUs. As illustrated in the table below, FedUni achieves approximately 19% higher accuracy compared to the second-best baseline, while maintaining comparable computation cost to many state-of-art approaches.
>
> |Run Time (s)|FedAvg|FedProx|Scaffold|Moon|FedSR|FedIIR|FedDG-GA|FedSDR|Ours|
> |-|-|-|-|-|-|-|-|-|-|
> |NVIDIA GeForce RTX 2080|1.33|1.87|1.94|2.04|2.12|1.95|3.05|5.02|3.26|
> |NVIDIA GeForce RTX 3090|0.77|0.92|0.91|1.13|1.53|0.86|1.40|2.60|1.57|
> |**Test Acc (%) （OOD-avg）**|68.53|68.23|68.47|73.92|72.69|68.99|69.02|68.72|**88.03**|
>
> In addition, we presented a brief proof for the convergence rate, showing that our convergence speed is also comparable to that of other methods.
>
> We assume that the objective functions $(F_{1}, \cdots, F_{N})$ satisfy L-smoothness and μ-strong convexity. The variances of the stochastic gradients are bounded, and the expected squared norms are also bounded.  Meanwhile, $(\Gamma=F-\sum_{k=1}^{N} p_{k} F_{k})$  is used to quantify the degree of non-iid data. In the case of full device participation, by choosing appropriate parameters $\kappa=\frac{L}{\mu}$, $\gamma = (\max{8\kappa, E})$ and the learning rate $\eta_{t}=\frac{2}{\mu(\gamma + t)}$. Our proposed algorithm satisfies $\mathbb{E}[F(w_{T})]-F^* \leq \frac{\kappa}{\gamma+T-1}(\frac{2B}{\mu}+\frac{\mu \gamma}{2} \mathbb{E}\| w_{1}-w^*| ^{2})$, where $B=\sum_{k=1}^{N} p_{k}^{2} \sigma_{k}^{2}+6 L \Gamma+8(E-1)^{2} G^{2}$, $\sigma_{k}^{2}$ represents the upper bound of the variance of the stochastic gradient in the k-th device,  E represents the number of local updates performed by each device between two communications. This indicates that when all devices participate and specific conditions are met, our proposed method can converge to the vicinity of the global optimal solution at a rate of $O(\frac{1}{T})$.
>
> **Q2: When to choose FedUni**
>
> The main advantage of FedUni is its ability to extract and retain all causal features from any input, making it well-suited for scenarios with **unknown test data distributions** and **client heterogeneity**.
>
> **Unknown test data distributions:** Our method is inherently **distribution-agnostic**, performing robustly in both ID and OOD settings. In contrast, existing approaches typically focus on either ID or OOD settings. ID-based methods inevitably exploit spurious correlations in the training data, undermining OOD generalization, while OOD-based methods usually discard client-specific knowledge, reducing ID discrimination ability. FedUni addresses both issues concurrently by incorporating a counterfactual generator to enhance OOD generalization and a feature compressor to preserve valuable client-specific information.
>
> **Client heterogeneity:** Our method is particularly effective in settings with data heterogeneity among clients, where existing approaches may discard valuable client-specific knowledge. Our method can preserve a broader set of causal features, leading to improved model performance.
>
> **Q3: Performance Gains in Homogeneous Setting**
>
> Reviewer pointed out that when training clients are similar, FedUni may not yield significant improvements. We conducted experiments in homogeneous setting (on the CMNIST dataset). FedUni still can outperforms other approaches. Whether to use FedUni in homogeneous setting depends on the trade-off between performance gain and computational cost.
>
> |Test Acc (%)|FedAvg|FedProx|Scaffold|Moon|FedSR|FedIIR|FedDG-GA|FedSDR|Ours|
> |-|-|-|-|-|-|-|-|-|-|
> |ID|97.08| 96.68| 97.06| 97.08|97.42|96.07|96.92|95.18|**98.16**|
> |OOD| 85.27| 85.46| 87.83|86.20|86.18|90.25|88.24|89.81|**97.30**|
>
> **Q4: More rigorous argument for Lemma 4.2**
>
> Thanks for your review! We will revise Lemma 4.2 into a more rigorous form in the final version.
>
> Inspired by ref [1], Lemma 4.2 relies on adopting specific $l$-norms to approximate the entropy terms in the coefficient of constraint. In particular,we rely on two main approximations.
> First, for the conditional entropy $H(x'|x)=-\mathbb{E}_X(\log P(x'|x)) \approx \mathbb{E}_X[\|\|x-\psi(x)\|\|_1]$,  where the approximation adopted in the last step amounts to assigning a $l_1$-Laplace distribution with identity covariance to the conditional probabilities:$P(x'|x)= \mathcal{L}(x';\mu(x), I)\propto \exp (-\|\|x'-x\|\|_1)$. Similiarily, for $H(x')$, we can derive $H(x')\approx\mathbb{E}_X[\|\|\psi(x)\|\|_1]$.
>
> [1] Pedro Savarese,et al. Information-Theoretic Segmentation by Inpainting Error Maximization. 2021.

---

### Decision · Program_Chairs · 2025-05-01

**Decision:**

Accept (poster)

**Comment:**

The reviewers generally agree that this paper merits acceptance; therefore, I recommend acceptance.

*Strengths.* The paper proposes a novel method that addresses the performance cost of only relying on invariant features when there might be other potentially casual features for some clients. The expected standard theoretical results are provided, and the empirical results cover reasonable datasets and baselines.

*Weaknesses.* The paper is quite dense. While the regularizers provided can be considered related to the desired properties, e.g., maintaining semantics, there is no provided guarantee or test of it.

The authors provided additional experiments to improve the empirical results during rebuttal and clarified reviewer questions. Most reviewers were positive before the rebuttal and remained positive after. I recommend acceptance conditioned on following through on the clarifications and discussions from the review process. Additionally, more discussion on privacy and the potential impacts of it in federated learning and in the context of the proposed method is neccesary.